# PELICAN: PERSONALIZED EDUCATION VIA LLM-POWERED COGNITIVE DIAGNOSIS AND ADAPTIVE TUTORING

## ABSTRACT

Personalized education aims to develop students' engagement, critical thinking and deep understanding through tailored teaching strategies. Although Large Language Models (LLMs) have generated significant attention in education due to their extensive knowledge base and reasoning capabilities, they still face challenges in personalized education, where the generation of correct, but one-size-fits-all responses fails to adapt to individual students' cognitive states, limiting their ability to support further heuristic learning. To address these challenges, we propose an adaptive tutoring framework consisting of two stages, integrating collaborative cognitive diagnosis with dynamic instructional adaptation. The first stage aims to model the student's cognitive state through collaborative cognitive diagnosis. The teacher utilizes a successor-first method to efficiently generate diagnostic questions, ensuring their accuracy through an expert-assistant-verifier pipeline. In the second stage, based on the estimated cognitive state, the teacher uses slow-thinking-based methods to select teaching strategies from a strategy pool to guide the student in solving problems. Teachers continuously monitor students' responses to provide feedback and track cognitive changes to ensure the most suitable strategies are used for effective tutoring. Evaluations on the Gaokao dataset demonstrate significant improvements in critical thinking stimulation (+18.7%) and task completion rates (+22.4%) compared to baseline models. Code is available at here.

## 1 INTRODUCTION

Personalized learning for individuals refers to an approach that adjusts the teaching process to meet the unique needs of each student. Unlike traditional one-size-fits-all methods, personalized learning ensures that instruction is flexible and responsive to a student's tutoring progress. This involves dynamically adapting pacing and strategies to better align with the student's cognitive level. The goal is not just individual mastery but also to foster active engagement, critical thinking, and deeper understanding.

However, directly adopting existing LLMs presents challenges in delivering individual-level personalized education. They often generate standardized responses(Tomisu et al., 2025), leading to two main issues (GUIDING). First, these responses lack differentiation, failing to account for dynamic variations in students' cognitive levels (Ippisch et al., 2025). Second, existing approaches prioritize correct problem-solving but overlook the need for personalized learning, resulting in a one-way information flow from teacher to student (Sonkar et al., 2024; Raja et al., 2025).

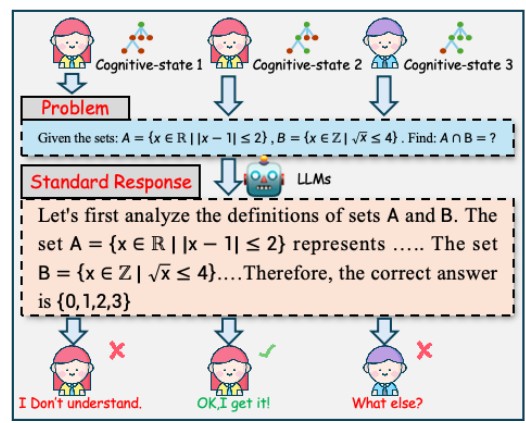

Figure 1: Standard responses generated by LLMs fail to meet individuals' diverse cognitive states, leading to poor learning results.

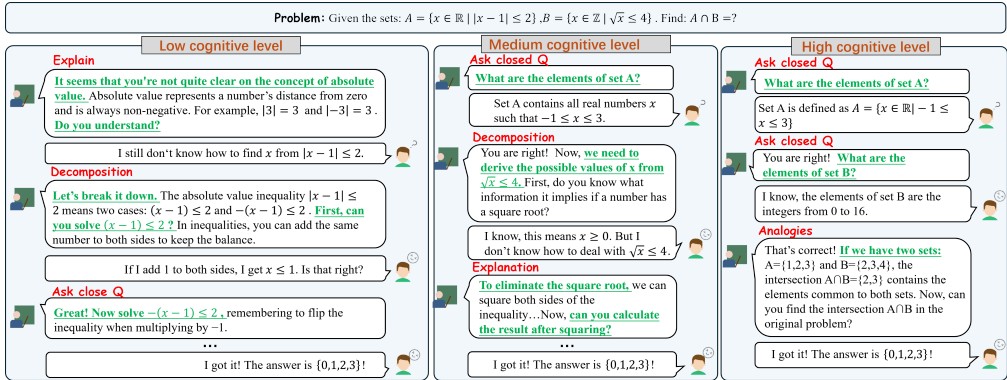

Figure 2: The teacher offer immediate feedback and select appropriate strategies to align with individuals' cognitive levels, thereby promoting problem-solving through active engagement.

As shown in Figure 1, the uniform responses generated by LLMs do not meet the diverse needs of students, who may passively absorb information that does not match their cognitive levels. To be effective, personalized education requires adaptive instruction that responds to students' evolving understanding, encouraging active problem-solving.

Although some studies have employed rule-based systems to model student types for targeted tutoring (Wang et al., 2024b), these approaches often fail to capture the complexity of individual differences and lack generalizability. Moreover, without incorporating cognitive modeling, purely heuristic methods cannot effectively adapt to students' unique needs. Other approaches, such as Socratic-style multi-turn dialogues, have aimed to encourage bidirectional communication (Ding et al., 2024; Liu et al., 2025; Planning). However, these methods often overlook the student's cognitive state, which can hinder engagement and task completion rates.

In recent years, large language models (LLMs) have shown significant potential in education due to their broad knowledge and advanced language comprehension abilities (Gonzalez et al., 2023; Jeon & Lee, 2023; Wang et al., 2024a; Huber et al., 2024; Wang & Demszky, 2023). However, despite these advancements, existing research largely overlooks the role of LLMs in personalized education, leaving a gap in their application for **tailoring learning experiences to individual students**. To overcome these challenges, we propose an adaptive tutoring framework consisting of two stages: a collaborative cognitive diagnosis stage followed by a dynamic tutoring stage. In the first stage, the teacher and student work together to assess the student's cognitive state, ensuring that the subsequent tutoring phase is tailored to the student's individual understanding. Based on this assessment, the second stage dynamically adjusts instructional support, fostering active problem-solving. Specifically, in the diagnostic phase, knowledge points are organized hierarchically, allowing the teacher to adjust the diagnostic sequence based on student performance and generate targeted questions to assess mastery, with an expert-assistant-verifier pipeline ensuring accuracy. In the tutoring phase, student responses are continuously monitored to provide feedback and track cognitive changes, enabling the teacher to refine tutoring strategies accordingly. Additionally, inspired by dual-system theory, teachers activate slow thinking when students face persistent cognitive challenges, simulating dialogue paths based on cognitive states to enhance engagement and improve completion rates. As shown in Figure 2, for students with varying cognitive levels, the teacher provides timely feedback on their needs and adapt strategies accordingly to match their cognitive capabilities, ultimately fostering problem-solving with active engagement. Our contributions are as follows:

- We introduce an adaptive tutoring framework that integrates collaborative cognitive diagnosis with adaptive tutoring, tailoring instructional support to students' cognitive levels and promoting active problem-solving.
- We employ the Slow-Thinking algorithm to select the most effective teaching strategies, simulating future dialogue paths to identify the best approach based on the student's cognitive state.
- We validate our method through experiments on the Gaokao dataset, demonstrating its effectiveness in assessing knowledge levels, stimulating critical thinking, and improving learning outcomes, highlighting its strong performance in personalized education.

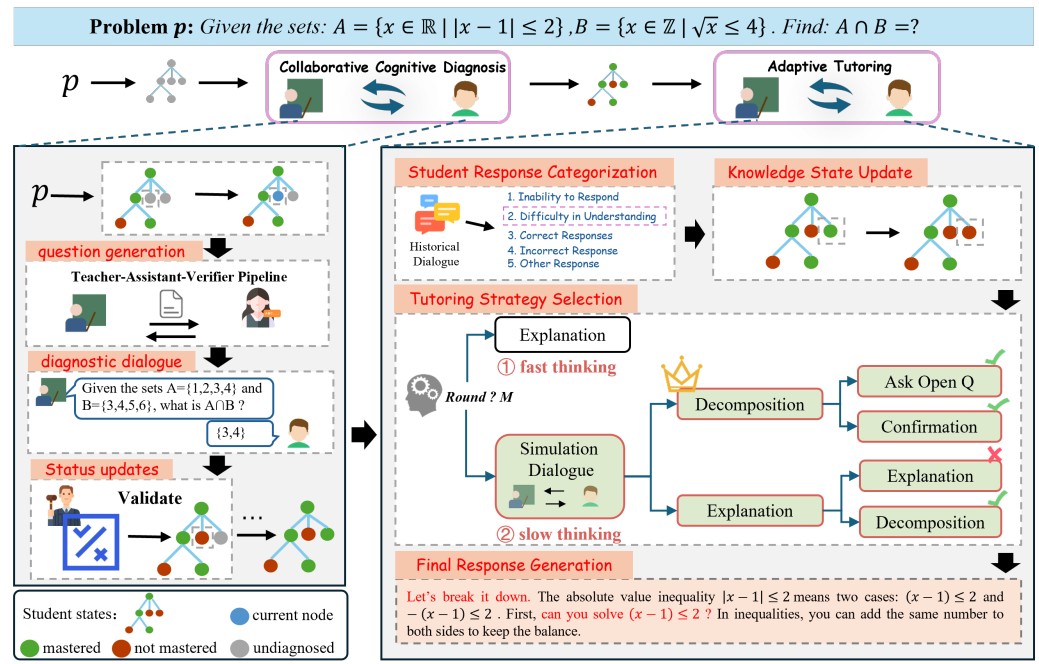

Figure 3: In the first stage, the teacher performs a cognitive diagnosis on the student in a successor-first order, with an expert-assistant-validator pipeline to ensure the correctness of the generated questions. This results in an estimated cognitive state, tailoring the subsequent tutoring to the student's understanding. In the second stage, student responses are monitored to provide feedback and adjust tutoring strategies as needed. When students face persistent cognitive challenges, slow thinking is activated by simulating dialogue paths based on cognitive states, aiming to enhance engagement and improve completion rates.

## 2 RELATED WORK

### 2.1 LLMS IN PERSONALIZED TUTORING

In recent years, LLMs have shown great potential in providing teaching support for students and teachers. For teachers, LLMs can generate teaching materials, design exercises, and assist in lesson preparation (Zheng et al., 2024; Caines et al., 2023; Gan et al., 2023). For student outcomes, LLMs analyze assignments and exams to provide assessments and feedback on learning progress, helping students optimize their learning paths (Dai et al., 2023). In terms of tutoring, some studies adopt Socratic-style heuristic teaching, encouraging students to think actively (Ding et al., 2024; Liu et al., 2025; Planning). Other research focuses on analyzing students' response types to provide appropriate feedback (Wang et al., 2024b). However, these methods overlook students' cognitive levels and individual differences, limiting their application in personalized education.

### 2.2 COGNITIVE DIAGNOSIS

Cognitive diagnosis aims to assess the mastery level students have achieved over specific knowledge concepts. Traditional methods, such as Item Response Theory (Fischer, 1995; Lord, 2012) and Multidimensional Item Response Theory (Adams et al., 1997; Ackerman, 2014), model students' proficiency using latent variables and logistic functions to predict their performance. However, these models do not take into account the specific knowledge concepts each item assesses or the relationships between them. In contrast, neural network-based models like NeuralCDM shift the focus to the relationships between students, items, and concepts (Su et al., 2024; Li et al., 2022b; Wang et al., 2022; Li et al., 2022a). While these models offer improvements, they still struggle with interpretability and do not account for the dynamic nature of cognitive development during tutoring.

## 3 Methods

### 3.1 Task Definition

The aim of this work is to provide personalized assistance to a student $u$ based on their cognitive levels, guiding them to actively solve problem $p$. The dialogue history at time step $t$, denoted as $D_t = \{q^1, r^1, \ldots, q^t, r^t\}$, consists of alternating teacher and student responses, where $q$ and $r$ represent the teacher's and student's responses, respectively. The process is divided into two stages. As shown in Figure 3, the first stage focuses on cognitive diagnosis to assess the student's knowledge level, and the second stage leverages this diagnosis to offer targeted tutoring.

**Collaborative Cognitive Diagnosis.** To map a student's cognitive state to specific knowledge points, we begin by extracting the relevant points from each problem. These points are organized into a hierarchical structure, where each parent node represents prerequisite knowledge (see Appendix B for details). A student can only master a child node after mastering its parent. The student's knowledge state $K_u$ for problem $p$ is then represented by binary values assigned to each node, indicating whether the student has mastered the corresponding knowledge point.

The goal of collaborative cognitive diagnosis is to assess the student's knowledge state through an interactive process. At step $t$, the teacher generates a question $q^t$ targeting a specific knowledge point, and the student's response $r^t$ is used to evaluate understanding. This iterative process produces an estimated knowledge state $\hat{K}_u$, which approximates the actual state $K_u$, and serves as the foundation for adaptive tutoring in the second stage.

**Adaptive Tutoring.** Once the knowledge state $\hat{K}_u$ is estimated, it is used to guide the student in solving problem $p$ via dialogue. Based on this dialogue history, the teacher generates a subsequent response $q^{t+1}$ to provide further support, continuing until the student successfully solves the problem.

### 3.2 Collaborative Cognitive Diagnosis

Traditional cognitive diagnostic methods primarily analyze student's interaction behaviors with historical exercises, but they often lack flexibility and interpretability (Wang et al., 2020; Su et al., 2024). In this paper, we propose a collaborative approach to diagnose a student's knowledge state $\hat{K}_u$, as shown on the left of Figure 3. Rather than evaluating each knowledge node independently, which can lead to inefficiencies and fragmentation, our method adjusts the diagnostic sequence dynamically based on the student's responses. Specifically, we introduce a successor-first strategy, in which the teacher prioritizes assessing leaf nodes or nodes whose successors have already been evaluated, leveraging the dependency structure of knowledge nodes. For each knowledge node $v$, the validator checks student's response $r^t$ to determine the value of $v$. If $v$ is mastered (value 1), all its prerequisite nodes are also updated to 1. This approach efficiently utilizes the hierarchical structure of knowledge to provide an accurate evaluation of the student's state.

The accuracy of the questions $q^t$ generated by the teacher is critical for reliable diagnostic assessments. A simple assumption is that if two experienced individuals provide the same answer to a question, the answer is likely correct. Based on this idea, we design an expert-assistant-verifier pipeline to ensure question correctness. In this process, the expert generates a question $q^t$ and its answer $a_r$, while an assistant language model independently answers the question, producing a response $a_s$. The verifier then compares $a_r$ and $a_s$ for consistency. If the answers do not match ($a_r \neq a_s$), the expert revises and regenerates the question until both answers align. This approach ensures the accuracy of diagnostic questions and minimizes errors.

### 3.3 Adaptive Tutoring

In this stage, we provide personalized tutoring based on the diagnostic results from the previous stage. This requires accurately assessing the student's knowledge state, delivering targeted feedback, and applying strategies to align their cognitive state, ultimately fostering individual mastery through active engagement. As shown on the right of Figure 3, to achieve this, we first capture the student's response state. Then we continuously update their cognitive state throughout the tutoring process, using a fast-slow thinking strategy selection algorithm based on dual-system theory, with the goal of aligning with the student's cognitive state and promoting active thinking.

To maintain coherence in the teacher's teaching approach and simulate a consistent tutoring style, inspired by Socratic principles (Liu et al., 2025), we decompose the problem $p$ into a series of step-by-step sub-tasks $\{sp_1, sp_2, \ldots, sp_n\}$. For each sub-task $sp_i$, the teacher provides personalized tutoring using the following steps.

### 3.3.1 STUDENT RESPONSE CATEGORIZATION

Effective teachers excel at categorizing student responses and providing feedback tailored to each one. We classify student responses into five categories: Difficulty in Understanding, where the student struggles to grasp a specific concept or process; Incorrect Response, where the student provides an incorrect answer; Inability to Respond, where the student cannot answer due to a knowledge gap; Correct Response, where the student provides the correct answer; and Other Responses, which encompass any responses that do not fall into the aforementioned categories. The teacher identifies the student's response type, denoted as $\text{type}^{(t)}$, which allows them to provide accurate and context-specific feedback.

### 3.3.2 KNOWLEDGE STATE UPDATE

As tutoring progresses, the student may gain new knowledge, leading to an update in their knowledge state. Additionally, misdiagnoses in the cognitive diagnosis process may occur and need to be corrected. Therefore, the teacher updates the estimated knowledge state $\hat{K}_u^{(t)}$ based on the student's response type $\text{type}^{(t)}$ and cognitive state $\hat{K}_u^{(t-1)}$ from the previous round.

### 3.3.3 TUTORING STRATEGY SELECTION

Students exhibit diverse cognitive levels and understanding abilities, which require teachers to choose the most appropriate strategies for each situation. Drawing from constructivist and scaffolding theories, we propose ten strategies that assist teachers in providing tailored support, which are detailed in Appendix E. However, the teacher often defaults to strategies like concept explanations or hints, which may not be sufficient when students face significant knowledge gaps. In these cases, teachers must make more precise choices.

The dual-system theory suggests two thinking modes: fast thinking, which is intuitive and rapid, and slow thinking, which is logical and deliberate. We apply this theory to strategy selection in teaching. As shown in the middle-left part of Figure 3, in the early stages of tutoring, the model uses "fast thinking" to select strategies based on the student's understanding:

$$c^{(t+1)} = \Phi_{\text{Gen\_c}}(Z, h^{(t)}, S) \tag{1}$$

where $Z = [p, sp_i, \hat{K}_u^{(t)}, \text{type}^{(t)}]$, $S$ denote the strategy pool and $h^{(t)}$ represents dialogue history. Fast thinking enables the teacher to quickly select an appropriate strategy based on experience.

When the number of dialogue rounds on a particular sub-task $sp_i$ exceeds a threshold $M$, which indicates that the student is likely encountering persistent cognitive obstacles. In this case, the LLM switches to slow thinking. Slow thinking identifies and evaluates promising strategies by simulating their potential teacher-student dialogue paths to select the optimal one. This is achieved by constructing a Simulated Teaching Tree, where each node represents a dialogue state and each action edge corresponds to a possible teaching strategy. Let $c^t$ denote the action taken by the teacher at the $t$-th dialogue turn, with the dialogue history defined as:

$$s^t = (c^1, q^1, r^1, \ldots, c^t, q^t, r^t). \tag{2}$$

Starting from the unique leaf node $s^t$, slow thinking searches for the next optimal action through an iterative planning process involving Simulated Teaching Tree **node expansion**, **dialogue simulation**, and **state evaluation** to update tree statistics. After $k$ iterations, it outputs the predicted next optimal action $c^{t+1}$.

**Node Expansion.** To balance selecting the most appropriate strategy with efficiency, the teacher generates the top $m$ candidate strategies for each leaf node $s^{tr}$ according to:

$$A = \Phi_{\text{Gen\_top}}(Z, s^{tr}, S) \tag{3}$$

where $A = \{\alpha_1, \alpha_2, \ldots, \alpha_m\}$ and $\alpha$ denotes the teaching strategy. This action broadens the teacher's strategic options, increasing the likelihood of selecting the optimal node.

**Dialogue Simulation.** The most effective way to measure the suitability of a strategy is by examining student feedback outcomes. Therefore, for each strategy, the teacher simulates a virtual tutoring session along with student responses, according to:

$$
\begin{aligned}
res_{tea}^{tr} &= \Phi_{\text{Sim\_T}}(Z, s^{tr}, \alpha) \\
res_{stu}^{tr} &= \Phi_{\text{Sim\_S}}(Z, s^{tr}, res_{tea}^{tr})
\end{aligned}
\tag{4}
$$

Thus, each of the $m$ strategies generates a new tree state $s^{tr'}$, resulting in $m$ new leaf nodes. The original state $s^{tr}$ is no longer considered a leaf.

**State Evaluation.** Each time a simulated dialogue is generated, teacher evaluates the strategy to determine whether the simulated tutoring is successful. Specifically, upon generating new state $s^{tr'}$, the teacher evaluates if the simulated student has mastered sub-task $sp_i$. If mastered, $s^{tr'}$ will be excluded from consideration in subsequent iterations; otherwise, simulation continues.

**Selection.** After all the simulations, we need to evaluate the value of candidate strategy $\alpha$. An intuitive approach is to assess all leaf nodes reached by executing $\alpha$ from state $s^t$. A successful leaf node suggests that the corresponding strategy is effective. Furthermore, fewer forward simulations indicate a more efficient strategy. Therefore, the score of a leaf node is defined as:

$$
\text{score} = 1 - \lambda \times (d - 1)
\tag{5}
$$

where $\lambda$ is a hyperparameter that penalizes deeper nodes, and $d$ represents the depth of the leaf node.

We calculate the score for the top $m$ strategies derived from state $s^t$, and the strategy with the highest total score is chosen as the optimal action $c^{(t+1)}$. Slow thinking not only takes into account the student's current cognitive state and understanding level but also anticipates the student's future learning state, ensuring the adaptability of the selected strategy. The integration of fast and slow thinking ensures that the decision-making process is both efficient and effective.

### 3.3.4 FINAL RESPONSE GENERATION

By utilizing the student's knowledge state $\hat{K}_u^{(t)}$, and the strategy $c^{(t+1)}$ employed in the current round, the teacher formulates the final response:

$$
q^{t+1} = \Phi_{\text{Gen\_res}}(Z, c^{(t+1)})
\tag{6}
$$

The response draws on the student's cognitive state, aiming to align with it and promote mastery through active engagement.

## 4 EXPERIMENTS

### 4.1 EXPERIMENT SETTINGS

**Datasets** We conduct experiments using the public Gaokao dataset[1], which includes 184 high school exam questions. Each question spans multiple knowledge points, with dependencies among them.

**Baselines** For cognitive diagnosis stage, we consider the following baselines: teacher-directed diagnosis methods (Free-Prompt, Cot), successor-first based diagnosis (No-Pipeline), and independent point-by-point diagnosis (S-Independent). For adaptive tutoring stage, we consider the following baselines: prompt-based tutoring approaches (Free-Prompt, Stepwise) and strategy-based approaches (Socratic (Liu et al., 2025), Bridge-Based (Wang et al., 2024b), Cot-bridge). Detailed descriptions are available in Appendix D.2.

**Metrics** For cognitive diagnosis stage, diagnostic quality is assessed using Precision, Recall, and F1-score (F1). Diagnostic efficiency is measured by the Average Diagnostic Round (Avg_Round), representing the number of rounds needed to assess a student's knowledge state. In the adaptive

---

[1] https://github.com/OpenLMLab/GAOKAO-Bench

tutoring stage, both automated and GPT-based assessments are used. Strict metrics, $R_{coverage}$ and $F_{frequency}$, are used to measure the proportion and frequency of non-mastered knowledge points addressed by the teacher, aiming to evaluate whether the teacher focuses on the student's cognitive state and non-mastered knowledge. For GPT-based assessments, five dimensions are considered: *Suitability*, *Logicality*, *Inspiration*, *Reliability*, and *Overall Quality*. More details can be found in Appendix D.3.

**Implementation Details** The assistant model in the first stage is GPT-3.5, while all other base models in both stages are GPT-4o (Achiam et al., 2023). In the second stage, slow thinking is activated after $M = 1$ rounds and iterates $k = 2$ to select the most appropriate strategy. For each leaf node, the number of generated candidate strategies is $m = 2$, and the penalty parameter for calculating the score is $\varphi = 0.4$. The maximum number of dialogue rounds for the baseline in the first stage is 15, while the second stage allows for 20 rounds of tutoring. In the main experiment, the slow-thinking process consumed ~230k tokens, accounting for ~40% of the total ~580k tokens used. Design details of the student role (Appendix G) and experimental costs (Appendix D.4) are provided.

## 4.2 MAIN RESULTS

**Collaborative Cognitive Diagnosis.** As shown in Table 1, PELICAN excels in diagnostic accuracy and efficiency, significantly outperforming the Free-Prompt and Cot methods due to its control over diagnostic sequencing and strategies. Meanwhile, the No-pipeline method, which omits the Expert-Assistant-Validator pipeline, results in less accurate diagnoses. Lastly, S-Independent, which diagnoses knowledge points independently, performs suboptimally compared to methods that account for dependencies between points.

**Adaptive Tutoring Stage.** As shown in Table 2, our method outperforms the baseline approaches significantly. By aligning the cognitive state, our method allows the teacher to focus more on the knowledge points the student has yet to master. Both Bridge-Based and Cot-Bridge directly select strategies, while our Slow-Thinking approach improves strategy appropriateness by constructing a Simulated Teaching Tree. Additionally, the explicit extraction of student state

Table 1: Performance of different Collaborative Cognitive Diagnosis methods. The best results are **bolded**.

| Methods | Precision↑ | Recall↑ | F1↑ | Avg_Round ↓ |
|---|---|---|---|---|
| Free-Prompt | 84.40 | 67.60 | 74.18 | 7.21 |
| Cot | 84.64 | 75.55 | 79.83 | 8.79 |
| No-Pipeline | 93.92 | 93.07 | 93.08 | 5.84 |
| S-Independent | 92.27 | 90.66 | 90.70 | 6.17 |
| PELICAN | **94.93** | **94.29** | **94.31** | **5.83** |

types and the systematic formulation of strategies further enhance performance across several metrics. We further provide the ANOVA analysis in Appendix K.1.

Table 2: Performance of different tutoring methods. PELICAN significantly outperforms the other methods, highlighting the effectiveness of student cognitive diagnosis-based tutoring.

| Methods | $R_{coverage}$ | $F_{frequency}$ | Suitability | Logic | Inspiration | Reliability | Overall |
|---|---|---|---|---|---|---|---|
| Free-Prompt | 59.81 | 59.75 | 2.91 | 4.18 | 2.42 | 3.66 | 3.60 |
| Stepwise | 58.17 | 59.61 | 4.2 | 3.99 | 3.96 | 3.64 | 3.80 |
| Socratic | 64.47 | 66.71 | 3.86 | 3.45 | 3.99 | 3.51 | 3.45 |
| Bridge-Based | 58.95 | 61.42 | 3.99 | **4.40** | 3.72 | 3.67 | 3.96 |
| Cot-Bridge | 56.29 | 57.06 | 3.69 | 4.23 | 3.26 | 3.59 | 3.68 |
| PELICAN | **72.36** ($\pm$4.69) | **72.06** ($\pm$3.42) | **4.27** ($\pm$0.003) | 4.37 ($\pm$0.014) | **4.21** ($\pm$0.002) | **4.51** ($\pm$0.006) | **4.33** ($\pm$0.003) |

## 4.3 ABLATION STUDIES

**Module ablation** We conduct ablation experiments on Collaborative Cognitive Diagnosis and slow-thinking modules. As shown in Table 3, the results indicate that removing any module degrades performance in $R_{coverage}$ and $F_{frequency}$, with the most significant drop occurring when both are absent, emphasizing the importance of these modules in addressing students' personalized needs. The absence of the slow-thinking module reduces the Suitability score, highlighting its role in strategy selection.

Table 3: Module ablation on stage-1 and slow-thinking. The results emphasize the key roles of both cognitive diagnosis and the slow thinking module in adapting to students' cognitive levels and promoting active thinking.

| Methods | $R_{coverage}$ | $F_{frequency}$ | Suitability | Logic | Inspiration | Reliability | Overall |
|---|---|---|---|---|---|---|---|
| *w/o.*Diagnosis | 47.76 | 48.21 | 4.22 | 4.10 | 4.48 | 3.78 | 4.24 |
| *w/o.*slow | 49.44 | 48.35 | 4.00 | 4.10 | 4.46 | 3.88 | 4.08 |
| *w/o.*Diagnosis & slow | 43.94 | 46.02 | 4.02 | 4.09 | **4.56** | 4.21 | 4.11 |
| PELICAN | **54.84** | **61.47** | **4.17** | **4.26** | 4.30 | **4.44** | **4.28** |

**Backbone Model Ablation** Table 4 presents the performance of PELICAN across various base models. All models perform well on the hard metrics, highlighting the significance of students' cognitive states in personalized tutoring. Notably, the GPT-4 model excels in suitability, logic, inspiration, and other areas, highlighting its superior language comprehension and ability to deliver personalized teaching support.

Table 4: Performance of PELICAN across various base models.

| Methods | $R_{coverage}$ | $F_{frequency}$ | Suitability | Logic | Inspiration | Reliability | Overall |
|---|---|---|---|---|---|---|---|
| LLama3.1-8b-Instruct | 54.79 | 55.7 | 2.46 | 2.84 | 3.18 | 2.98 | 2.9 |
| GLM-4-PLUS | 59.18 | 56.00 | 4.10 | 4.12 | 4.24 | 4.18 | 4.06 |
| Qwen-max | **64.41** | 58.07 | **4.20** | 3.96 | 4.28 | 4.02 | 3.92 |
| Ours(GPT-4o) | 54.84 | **61.47** | 4.17 | **4.26** | **4.30** | **4.44** | **4.28** |

## 4.4 IN-DEPTH ANALYSIS

**Student Cognitive Levels Analysis** We evaluate the performance of PELICAN in tutoring students at different cognitive levels. Specifically, for each math question in the Gaokao dataset, we initialize three different cognitive levels for the students: low, medium, and high, details see Appendix G.3. Table 5 presents the success rate (SR) and average tutoring rounds (ADR) for these cognitive levels. The results show that as students' abilities increase, their success rates improve, and the number of tutoring rounds decreases. Notably, even students with the lowest cognitive level achieve a success rate of 75%, with only a 7.5% gap compared to the highest-level students. This demonstrates that PELICAN can effectively support students across a wide range of cognitive levels.

**Strategy Adaptation by Cognitive Level** We also present the strategy distribution used to tutor students at different cognitive levels. As shown in Figure 4, *explaining* is the most commonly used strategy by teachers across all levels, emphasizing its importance in facilitating understanding at every stage of learning. For higher-level students, teachers tend to use *questioning* strategies than with students at other cognitive levels. For lower-level students, *providing analogies* is a frequently employed strategy. This is because students with lower cognitive levels often struggle to grasp abstract concepts and analogies help make these concepts more concrete and relatable. More experimental results are presented in Appendix K.

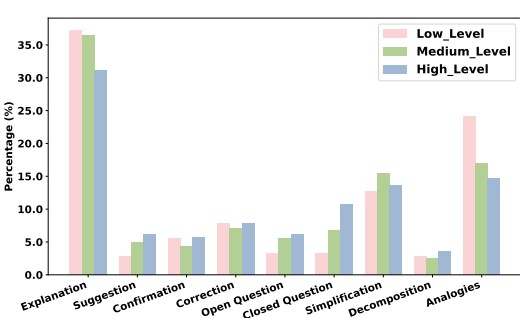

Figure 4: The strategy distribution used to tutor students at different cognitive levels.

## 4.5 CASE STUDY

In Figure 5, **PELICAN** extracts knowledge points and conducts a dialogical cognitive diagnosis, ultimately determining the student's cognitive state $\hat{K}_u$. In the second stage, the teacher analyzes the student's response type and cognitive level, identifying areas of confusion in the student's understanding of the definition of even functions. To address this, the teacher provides examples and

Figure 5: PELICAN generates personalized responses through cognitive diagnosis and dynamic tutoring. It identifies the student's misconception of the dual function, simplifies the concept with examples, and encourages active computation.

encourages the student to actively engage in the steps for determining even functions. This approach aligns with the student's cognitive level and fosters active thinking. The **Free-Prompt** Method explains the concept and provides an example of determining even functions, followed by solving the problem steps. While easy to understand, this method lacks the promotion of active student engagement. For **Sepwise**, the teacher directly asks questions on specific knowledge points based on the problem-solving steps, without considering the student's cognitive state or needs, resulting in higher difficulty for the student. The **Socratic** method explains the definition of even functions, but it's difficult for students with lower cognitive levels to grasp, leading to ineffective tutoring.

### 4.6 HUMAN EVALUATION

We further conducted a real-world experiment involving 169 high school students, collecting a total of 1335 tutoring reports, with each student submitting an average of 7.90 reports. Detailed experimental procedures, ethical considerations, and ANOVA analysis can be found in the Appendix I. As shown in Table 6, the results exhibit strong consistency with the GPT-based evaluation outcomes presented in Table 2. Our method demonstrates a high success rate, confirming the effectiveness of the tutoring approach.

Table 5: Performance of PELICAN in tutoring students at different cognitive levels.

| Cognitive Level | Low | Medium | High |
|---|---|---|---|
| SR | 75.00 | 80.00 | 82.50 |
| ADR | 9.00 | 8.10 | 6.97 |

Table 6: Results from a real-world student experiment.

| | Success rate | $R_{coverage}$ | $F_{frequency}$ | Appropriateness | Sentiment | Inspiration | Overall |
|---|---|---|---|---|---|---|---|
| Free-Prompt | 85.2% | 60.71 | 60.76 | 3.05 | 3.06 | 1.98 | 2.35 |
| Stepwise | 86.5% | 58.53 | 58.60 | 3.79 | 2.67 | 3.76 | 3.57 |
| Socratic | 80.3% | 63.91 | 63.88 | 3.85 | 3.62 | 4.01 | 3.53 |
| Bridge-Based | 80.1% | 58.73 | 58.69 | 4.02 | 4.11 | 3.93 | 4.10 |
| Cot-Bridge | 82.8% | 57.72 | 57.65 | 4.07 | 3.97 | 3.97 | 4.14 |
| PELICAN | **86.8%** | **70.04** | **70.07** | **4.23** | **4.42** | **4.33** | **4.39** |

## 5 CONCLUSION

In conclusion, this paper introduced PELICAN, a novel adaptive tutoring framework that combines collaborative cognitive diagnosis with dynamic instructional adaptation to personalize education. Our approach effectively tailors teaching strategies to individual cognitive states, fostering deeper engagement and critical thinking. By leveraging a hierarchical knowledge structure for fine-grained cognitive diagnosis and incorporating a dual-system theory (Kahneman, 2011) for strategy selection, PELICAN ensures that tutoring adapts to both current and future learning needs. Extensive experiments conducted on the Gaokao dataset highlight the effectiveness of PELICAN in addressing the diverse, dynamic needs of students across various subjects and cognitive states, offering a more personalized and effective learning experience.

ETHICS STATEMENT

This research strictly adheres to ethical guidelines for responsible AI and machine learning research. Since the participants were high school students, informed consent forms were distributed to parents or legal guardians of all participating students through school channels prior to the experiment, and their written signatures were obtained. Simultaneously, the study's purpose and procedures were explained to the students themselves, and their written assent was acquired.

During the experiment, we strictly complied with regulations for the protection of minors, ensuring voluntary participation, anonymization of personal information, and the constant presence of teachers and school staff throughout all learning and testing activities to safeguard the students' well-being and data security. All collected data were anonymized and used exclusively for research purposes. The study was reviewed and approved under our institution's internal review procedures.

We acknowledge that responsible research necessitates careful consideration of potential harms and social impacts. Our work does not involve manipulative, discriminatory, or unsafe practices, and our experimental design was consistently guided by the principle of minimizing negative consequences for participants and the broader community.

REPRODUCIBILITY STATEMENT

To ensure the reproducibility of our research, we have implemented the following measures: First, the main text provides comprehensive descriptions of our methodological models, and the corresponding code has been made publicly available via an anonymous repository (with access details provided in the supplementary materials). Second, all datasets used in our experiments are publicly accessible, and the anonymous repository also includes complete data processing workflows along with step-by-step guidelines for experimental replication. Finally, the appendix contains detailed specifications of the experimental setup, hyperparameters, and evaluation protocols to enable precise replication of both computational and human-in-the-loop experiments. Collectively, these resources form a complete infrastructure necessary for reproducing the results reported in this study.

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

## A  STATEMENT ON THE USE OF LLMS

In the preparation of this manuscript, we utilized a large language model. Specifically, GPT-4V was employed to assist with grammar checking and text polishing, thereby enhancing the accuracy and readability of the content. It is important to note that the authors assume full responsibility for all material presented herein. The final version has been carefully reviewed and reflects the authors' own academic perspectives.

## B  CONSTRUCTION OF KNOWLEDGE STATE TREE

**Construction** To enhance the accuracy of the knowledge tree, we chose a model-based approach for node selection instead of direct generation. The process is as follows: **(1)Manual Tree Construction**: We first manually build comprehensive knowledge trees for four subjects based on high school exam syllabi. These trees are then reviewed and refined by education experts to ensure accuracy and relevance. **(2)Subtree Generation for Each Question**: For each question, we generate a subtree from the full tree. The model begins by identifying the relevant subject and selecting the primary knowledge points (e.g., sets, algebra). After pinpointing the primary knowledge point, the model selects secondary knowledge points (e.g., plane geometry, solid geometry). This step-by-step process simplifies the task of node selection.

Each node in the tree contains no more than 20 child nodes, making it easier for the model to select the appropriate one. Moreover, we provide both the question and its explanation, which reduces the

difficulty of node selection. The overall time complexity of the construction process is $O(n \log n)$, ensuring high efficiency. The final depth of each question's knowledge tree spans 2 to 4 layers. This approach ensures that the generated tree is a valid subtree of the syllabus, aligning with real-world educational structures.

**Validation** We introduced manual evaluation to further assess the constructed trees. Through manual checks and corrections, we found errors in fewer than 3% of cases (less than 5 questions). Consequently, the manpower cost is minimal.

**Examples** The following are specific examples from the constructed knowledge trees. These trees will be made available on the project homepage in the future:

**Example 1: Inheritance Hierarchy**

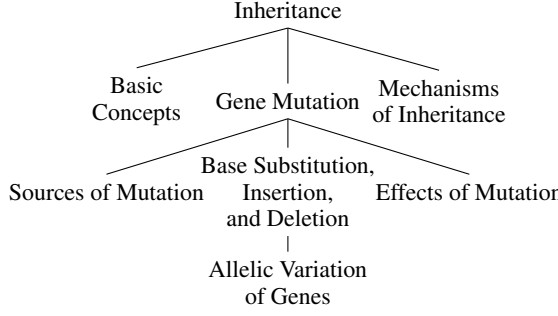

**Example 2: Complex Numbers Concepts**

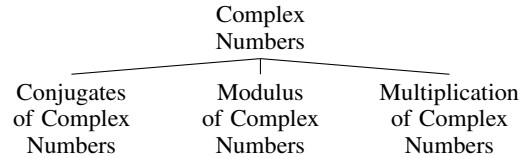

**Example 3: Organic Chemistry Isomers**

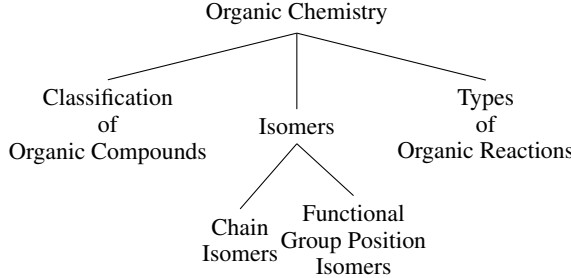

## C DIFFERENCE FROM COGNITIVE STATE TRACKING

**Modeling Approaches** Existing models typically rely on either learning history and current goals [1] or skill mastery to generate instructional materials, though they lack a unified standard and consistent granularity [2]. In contrast, our approach models knowledge using a knowledge state tree, derived from the college entrance examination syllabus, which defines the hierarchical relationships between

knowledge points. We use an expert-assistant-verifier pipeline to generate diagnostic questions, offering a more precise and theoretically grounded method for assessing knowledge state mastery.

**Response Handling** Traditional response handling is reactive, only addressing learner queries without offering proactive guidance [1]. Some systems generate feedback based on learner interactions, but the feedback lacks detailed categorization [2]. Our approach, however, classifies learner responses in more detail—such as identifying understanding difficulties, incorrect answers, or the inability to respond. This classification enables dynamic updates to the learner's cognitive state, allowing for the identification of specific obstacles and the provision of targeted feedback. The system can then adjust teaching strategies, improving both the relevance and effectiveness of the tutoring process.

**Tracking Cycles:** In existing systems, tracking typically occurs after each learning session to generate a course plan [1] or to update content based on skill mastery [2]. However, these methods fail to promptly identify students' learning states and provide timely feedback. In contrast, our approach updates the student's cognitive state after each tutoring turn and employs slow thinking to select strategies that align with the learner's current cognitive level. This ensures the tutoring content is relevant, thereby boosting learning efficiency and motivation.

We clarify that cognitive state tracking is not the primary focus of this paper, but rather a tool to provide targeted tutoring that responds to students' specific needs.

## D   EVALUATION SETTINGS

### D.1   DATASETS

The Gaokao dataset used in our experiment consists of 184 high school exam questions across mathematics, biology, chemistry, and physics. Each question encompasses multiple knowledge points, with identifiable dependencies between them, reflecting a sequential relationship.

### D.2   BASELINES

#### D.2.1   BASELINES IN STAGE-1

To evaluate the effectiveness of our proposed method, we design four baseline models with different diagnostic strategies for comparison.

**Free-Prompt** Teachers conduct multi-turn dialogues for knowledge point diagnosis based solely on prompts. The diagnostic order and strategy are entirely determined by the teacher.

**Chain-of-Thought Diagnosis (Cot)** This baseline incorporates Chain-of-Thought reasoning into the diagnostic process. Before generating each response, the teacher explicitly considers which knowledge points have been diagnosed, which remain, and which should be diagnosed in the current turn.

**Pipeline-Free Diagnosis (No-Pipeline)** This variant follows the same diagnostic framework as our approach but removes the Expert-Assistant-Validator pipeline.

**Sequentially Independent Diagnosis (S-Independent)** This variant follows the same diagnostic framework as our approach but all knowledge points are treated independently, disregarding their interdependencies. The teacher diagnoses each knowledge point separately.

#### D.2.2   BASELINES IN STAGE-2

**Prompt-Based Tutoring (Free-Prompt)** The teacher directly tutors the student using a multi-turn dialogue based on a single prompt.

**Stepwise Prompt-Based Tutoring (Stepwise)** This method improves upon **Free-Prompt** by explicitly decomposing the problem into multiple steps. The teacher provides tutoring following these predefined steps.

**Socratic Tutoring (Socratic)** This baseline engages students through heuristic questioning, guiding them step by step toward self-discovery rather than providing direct answers.

**Bridge-Based Tutoring (Bridge-Based)** Bridge follows a structured decision-making process, where the tutoring is divided into three stages: (1) extracting the student's response type, (2) selecting an appropriate strategy, and (3) generating the final response. We adapt Bridge's strategies to better fit the tutoring task.

**Chain-of-Thought Bridge-Based Tutoring (Cot-Bridge)** Building upon bridge, this variant incorporates a Chain-of-Thought reasoning mechanism, where the teacher explicitly determines the student's response type, tutoring strategy, and intent before generating a response.

### D.3 METRICS

The evaluation of Stage 2 combines both automated and GPT-based assessments. We use strict metrics to evaluate whether teachers focus on students' non-mastered knowledge points. The metric $R_{coverage}$ measures the proportion of unique non-mastered knowledge points (where the value is 0) out of all the knowledge points explained by the teacher:

$$R_{coverage} = \frac{|\{p_i \mid \text{value}(p_i) = 0, p_i \in \mathcal{P}\}|}{|\mathcal{P}|}$$

Here, $\mathcal{P}$ represents the set of all knowledge points explained by the teacher, $\text{value}(p_i)$ denotes the mastery status of knowledge point $p_i$ (where 0 indicates a non-mastered point), and $p_i$ is a specific knowledge point. The metric $F_{frequency}$ measures how often non-mastered points (value = 0) are addressed, accounting for repeated explanations:

$$F_{frequency} = \frac{\sum_{p_i \in \mathcal{P}, \text{value}(p_i)=0} \text{count}(p_i)}{\sum_{p_i \in \mathcal{P}} \text{count}(p_i)}$$

Here, $\text{count}(p_i)$ refers to how many times a knowledge point $p_i$ is explained, and the sum in the numerator accounts for repeated explanations of non-mastered points. For GPT-based assessments, five dimensions are considered:**Suitability**: Suitability of the tutoring method. **Logicality**: Coherence and consistency of explanations. **Informativeness**: Depth and insights provided. **Reliability**: Consistency of the tutoring approach. **Overall Quality**: General quality of the session.

### D.4 SCALE EFFECTS OF THE SLOW THINKING

We conduct experiments on the parameters in slow thinking, considering the impact of the number of iterations $k$ and the number of candidate strategies $m$ on the experimental results. Results are shown in Table 9. The findings reveal that the k=2, m=2 configuration achieves the best performance. The slow-thinking approach achieves substantial performance gains across multiple metrics while requiring only minimal simulations.

The slow-thinking mode is used infrequently, with only 14.7% of conversations triggering it. In the main experiment, 184 questions were tutored, and the total slow-thinking overhead amounted to 11.47 dollars, out of a total cost of 28.91 dollars.

In practical applications, tutoring is typically divided into independent steps. Therefore, during simulation, we can reduce the overhead by inputting only the most recent rounds of conversation history, which further minimizes the token cost of the input context.

## E STRATEGY DETAILS

The paper CIMA (Stasaski et al., 2020) proposes four basic teaching strategies: Hint, Open-Ended Question, Correction, and Confirmation.

Inspired by **constructivist theory** (Hein, 1991) (which emphasizes building new knowledge upon existing understanding) and **scaffolding theory** (Van Der Stuyf, 2002) (which advocates providing adaptive support to develop learner independence), we supplemented these with additional strategies: Explanation to clarify concepts, Simplification to reduce complexity, Decomposition to break down problems, and Analogy to bridge abstract ideas with concrete examples.

This final strategy pool not only covers essential teaching scenarios from concept introduction to error correction, but also more effectively promotes students' active cognitive engagement.

Table 7 presents the specific details of the teaching strategy.

| Strategy Label | Description |
|---|---|
| Explain a concept (Explanation) | Explain a concept or method to the student without directly providing the answer to the question. |
| Suggest a strategy (Suggestion) | Guide the student to adopt a particular method or strategy to encourage further thinking. |
| Confirm the student's answer (Confirmation) | Affirm the student's correct response and encourage them to continue thinking about the next step. |
| Correct the student's answer (Correction) | Gently point out the student's mistake and guide them to reassess the problem for self-correction. |
| Ask an open-ended question (Open Question) | Prompt the student to think deeply or have the student complete the next step of the answer. |
| Ask a closed-ended question (Closed Question) | Test the student's understanding and memory, focusing attention on specific knowledge points. |
| Simplify the question (Simplification) | Break the question down into a simpler form to help the student grasp the core idea. |
| Decompose the question (Decomposition) | Break the question into smaller, more manageable parts to help the student solve it step by step. |
| Provide analogies or examples (Analogies) | Use examples or analogies from everyday life to help the student better understand abstract concepts. |
| Other | Any response that doesn't fit the above categories. |

Table 7: Strategy Details

## F    SLOW-THINKING ALGORITHM

Algorithm 1 show the details of Slow-thinking Algorithm.

---
**Algorithm 1** Slow Thinking for Teaching Strategy Optimization

---
**Input**: Initial state $s_t$, sub-task $sp_i$, number of iterations $k$, parameter $\lambda$
**Output**: Optimal action $c^{t+1}$
1: $Leafs \leftarrow [s_t], S\_leafs, N\_leafs \leftarrow [], []$
2: **for** $iter = 1$ to $k$ **do**
3:      **for** $s^{tr} \in Leafs$ **do**
4:          $A \leftarrow \Phi_{\text{Gen\_top}}(Z, s^{tr}, S)$
5:          **for** $\alpha \in A$ **do**
6:             $res_{tea}^{tr} \leftarrow \Phi_{\text{Sim\_T}}(Z, s^{tr}, \alpha)$
7:             $res_{stu}^{tr} \leftarrow \Phi_{\text{Sim\_S}}(Z, s^{tr}, res_{tea}^{tr})$
8:             $s^{tr'} \leftarrow (s^{tr}, res_{tea}^{tr}, res_{stu}^{tr})$
9:             Evaluate $s^{tr'}$
10:            **if** success **then**
11:               $S\_leafs.append(s^{tr'})$
12:            **else**
13:               $N\_leafs.append(s^{tr'})$
14:            **end if**
15:          **end for**
16:      **end for**
17:      $Leafs \leftarrow N\_leafs$, Clear $N\_leafs$
18: **end for**
19: Initialize scores dictionary $Scores \leftarrow \{\}$
20: **for** $\alpha$ in initial top strategies generated from $s_t$ **do**
21:      $Scores[\alpha] \leftarrow \sum_{node \in Leafs \cup S\_leafs, node.strategy == \alpha}(1 - \lambda \times (d - 1))$
22: **end for**
23: Select optimal action based on highest score $c^{t+1} \leftarrow \arg\max_{\alpha}(Scores[\alpha])$

---

# G  STUDENT SIMULATION

## G.1  INITIALIZATION OF THE STUDENT'S COGNITIVE STATE

The student is played by GPT-4o. For each problem $p$ and its solution $s$, we extract the necessary knowledge points and construct a hierarchical tree, where each parent node represents prerequisite knowledge. The student's knowledge state $K_u$ for problem $p$ is then determined by assigning binary values to the nodes in a top-down manner, indicating whether each knowledge point has been mastered. Specifically, the topmost node is always initialized to 1. Then, starting from the top, the nodes are randomly initialized: if a parent node is 0, its child node must be initialized as 0; if a parent node is 1, its child node is randomly initialized to either 0 or 1. This process ultimately results in the student's cognitive state $K_u$.

## G.2  STUDENT RESPONSE GENERATION

**Stage-1**

At this stage, for the teacher's questions, the LLM, acting as the student, generates responses based on the provided cognitive state. The model is prompted to ensure that the generated responses are consistent with the cognitive state.

**Stage-2**

In this stage, student's response generation follows a structured process. First, the teacher's questions are categorized into three types:

- **Type 1:** First explain the knowledge, then ask a question.
- **Type 2:** Directly ask a question.
- **Type 3:** Other.

The student's responses are categorized into five types:

- **Type 1**: Does not understand and asks a question.
- **Type 2**: Provides an incorrect answer.
- **Type 3**: Unable to answer the question.
- **Type 4:** Provides the correct answer.
- **Type 5:** Other.

We use a "first classify, then randomize" strategy to determine the student's response type for each round. We classify the teacher's last interaction as follows:

- If the teacher first explains and then asks a question (Type 1), the student randomly selects an answer from Type 1, 3, or 4.
- If the teacher directly asks a question (Type 2), the model assesses the student's knowledge level to determine if they can answer correctly (Type 4). If they cannot, the response will be randomly selected from Type 2 or 3.
- If the teacher's question belongs to Type 3, the student's response will be classified as Type 5.

Additionally, to simulate the changes in the student's cognitive state throughout the learning process, we have designed a state updater based on the LLM. After each round, the student's knowledge is adjusted. Specifically, if the student answers correctly (Type 4), the cognitive state is updated to reflect the mastery of new knowledge. The system will provide the correct answer to the current sub-step, ensuring the accuracy of the student's response, particularly for Type 4.

The final student response is based on the dialogue history, the response type for the current round, and the cognitive state, generating a response that aligns with the cognitive state.

## G.3  INITIALIZATION STUDENT STATE METHODS IN STUDENT COGNITIVE LEVELS ANALYSIS

For the experiment, we initialize the student states in the following manner:

- **Low Cognitive Level**: The student has a mastery level of 1 only at the root node, with all other knowledge points initialized to a mastery level of 0.

- **Medium Cognitive Level**: One leaf node is randomly initialized with a mastery level of 0, and all other nodes are set to 0.

- **High Cognitive Level**: The student state is randomly initialized in a top-down manner, considering the dependency relationships between knowledge points.

## H   FAILURE SCENARIOS ANALYSIS

In problems that demand high levels of comprehension and spatial imagination, such as geometric problems in mathematics or isomer calculations in chemistry, students often struggle to grasp abstract concepts when relying solely on text-based explanations. Even when strategies like analogies, examples, and problem breakdowns are employed, these challenges often lead to suboptimal tutoring outcomes.

Consider the following geometry problem: A triangular prism has side edges perpendicular to its base, all edges of length $a$, and its vertices lie on the surface of a sphere. What is the surface area of the sphere? A. $\pi a^2$ B. $\frac{7}{3}\pi a^2$ C. $\frac{11}{3}\pi a^2$ D. $5\pi a^2$

Despite the teacher's attempt to explain through analogy: "Let's explore why the vertices of an equilateral triangle lie on the same spherical surface. Imagine ... Can you visualize the relationship between the prism and the sphere? It's like a tent standing inside a sphere." The student still struggle to visualize the spatial relationship between the triangular prism and the sphere, resulting in difficulties in understanding the problem.

In such cases, relying solely on textual descriptions and analogies clearly fails to effectively help students build spatial visualization. The ideal teaching method should incorporate diagrams or dynamic visualizations to help students form an intuitive understanding. Consequently, the failure to achieve desired tutoring outcomes is common when dealing with such high-difficulty spatial problems. In future work, we plan to integrate multimodal generation to provide more effective support for spatial visualization.

## I   REAL-WORLD STUDENT EXPERIMENT

### I.1   ETHICAL CONSIDERATIONS

Since the participants were high school students, we distributed informed consent forms to the parents or legal guardians of all participating students through the school channels before the experiment, and obtained their written signatures. At the same time, we also explained the purpose and procedures of the study to the students themselves, and obtained their written assent.

During the experiment, we strictly followed the regulations for the protection of minors, ensuring voluntary participation, anonymization of personal information, and the presence of teachers and school staff throughout the entire learning and testing process to safeguard the students' well-being and data security.

### I.2   EXPERIMENT DETAILS

The experiment was conducted in the following steps: **(1)** For each participant, we randomly selected 30 questions from a pool of 184, and participants chose all the questions they did not know how to answer; **(2)** Participants received tutoring from the model; **(3)** Participants completed a feedback evaluation, first selecting whether they had mastered the question, followed by rating the tutoring process on four dimensions: appropriateness, emotional experience, inspiration, and overall experience, using a scale from 1 (lowest) to 5 (highest).

In step (1), the number of questions selected by the students can somewhat reflect their cognitive level. The student who selected the most questions chose 15, while the student who selected the fewest chose 2.

In step (2), we compare our method with five baseline approaches. For each question, one of six tutoring methods is randomly selected (with equal probability) to tutor the student, with up to 20 rounds per question.

## I.3 POWER ANALYSIS

Assuming an effect size of 0.5, $\alpha = 0.05$, power = 0.8, and six experimental groups, the required sample size was 148. Therefore, we recruited a total of 169 students to participate in the experiment. These participants collectively contributed 1335 experimental reports, with an average of 7.90 reports per person.

We conducted one-way ANOVA across the six tutoring conditions for all subjective evaluation metrics in Table 8.

Table 8: One-way ANOVA results for subjective evaluation metrics

| Metric | F-value | p-value | Conclusion |
|---|---|---|---|
| Appropriateness | 3.43 | 0.039 | Significant main effect |
| Sentiment | 3.17 | 0.043 | Significant main effect |
| Inspiration | 2.96 | 0.027 | Significant main effect |
| Overall | 3.58 | 0.035 | Significant main effect |

All p-values were below 0.05, confirming significant differences among the six tutoring methods on subjective experience and learning outcomes.

## I.4 STUDENT FEEDBACK

At the end of the experiment, students completed a satisfaction survey and highlighted the most helpful dialogue turns. Feedback showed benefits beyond task performance:

- 34% felt more at ease interacting with the virtual teacher;
- 27% found the guidance targeted;
- Among low-performing students, 39

Students marked 13% of key turns in the fast-thinking phase and 24% in the slow-thinking phase, indicating stronger preference for the slow-thinking phase.

## J REAL-WORLD DEPLOYMENT

In this paper, Table 4 presents the performance of our method across different models, while Table 11 further examines its performance on models of varying sizes. Specifically, Qwen2.5-32b-instruct ($0.0003 per thousand tokens) achieves results comparable to GPT-4o.

Additionally, as the model size decreases, our method continues to perform well on Qwen2.5-7b-instruct and Qwen2.5-13b-instruct, demonstrating that our approach effectively balances cost and performance, offering strong scalability.

Furthermore, the model can be deployed via cloud servers and accessed through API calls, so it does not place significant demands on users' client-side resources. Thanks to the flexibility of cloud services, computational resources can be dynamically allocated during periods of high demand, enabling elastic deployment to accommodate fluctuations in usage.

## K MORE EXPERIMENTAL RESULTS

**Model Scale Ablation**

Table 9: Slow-thinking ablation

|  | Suitability | Logic | Inspiration | Reliability | Overall | $R_{coverage}$ | $F_{frequency}$ |
|---|---|---|---|---|---|---|---|
| no-slow-thinking | 4.00 | 4.10 | **4.46** | 3.88 | 4.08 | 49.44 | 48.35 |
| m2k1 | 4.18 | 4.16 | 4.22 | 4.26 | 4.06 | **54.95** | 48.80 |
| m2k2 | 4.17 | 4.26 | 4.30 | **4.44** | **4.28** | 54.84 | **61.47** |
| m2k3 | **4.28** | 4.34 | 4.22 | 4.06 | 4.00 | 47.01 | 54.89 |
| m3k1 | 4.08 | 4.06 | 4.22 | 4.30 | 3.98 | 51.02 | 47.36 |
| m3k2 | 4.22 | **4.36** | 4.02 | 4.24 | 4.18 | 53.64 | 57.79 |

Table 11: The impact of different model scales on the experimental results.

| Methods | Suitability | Logic | Inspiration | Reliability | Overall | $R_{coverage}$ | $F_{frequency}$ |
|---|---|---|---|---|---|---|---|
| Qwen-max | 4.2 | 3.96 | 4.28 | 4.02 | 3.92 | **64.41** | 58.07 |
| Qwen2.5-7b-instruct | 3.78 | 3.6 | 4.48 | 3.22 | 3.70 | 42.59 | 42.82 |
| Qwen2.5-13b-instruct | **4.46** | 4.24 | **4.62** | 4.24 | 4.30 | 44.44 | 50 |
| Qwen2.5-32b-instruct | 4.34 | **4.32** | 4.54 | 4.12 | **4.36** | 44.86 | 47.92 |
| Ours(GPT-4o) | 4.17 | 4.26 | 4.3 | 4.44 | 4.28 | 54.84 | **61.47** |

Table 11 presents the impact of different model scales on the experimental results. The experiment evaluates the Qwen-max(Bai et al., 2023) and Qwen2.5Instruct series across models of 7b, 13b, and 32b sizes. The instruction-tuned models (Qwen2.5-13b-instruct, Qwen2.5-32b-

Table 10: Effectiveness of Expert-Assistant-Verifier pipeline

|  | Turn=1 | Turn=2 | Turn=3 |
|---|---|---|---|
| Accuracy | 88.6% | 93.7% | 94.5% |

instruct) outperform the base models (Qwen-max, GPT-4o), demonstrating the benefits of instruction fine-tuning in improving instruction execution. The Qwen2.5Instruct-7b model shows the weakest performance, while the 13b and 32b models yield similar results, suggesting that when the model parameters are small, model size is the primary factor limiting performance.

**Ablation of Base Model** Table 4 presents the performance of PELICAN across various base models. The GPT-4 model excels in suitability, logic, inspiration, and other areas, highlighting its superior language comprehension and ability to deliver personalized teaching support.

**Effectiveness of the Expert-Assistant-Verifier pipeline**

For the 184 high school entrance examination questions, we generated a total of 1,260 questions for cognitive diagnosis, with an average of 1.24 rounds of generation per question. We manually verified the correctness of each question. In the table 10, we investigate the relationship between the accuracy of the generated questions and the maximum number of executions in the Expert-assistant-verifier pipeline. When Turn = 1, no pipeline is executed. The results show that the accuracy of the questions improves as the maximum number of executions increases, highlighting the effectiveness of our pipeline design.

**Generalizability of the Method** To validate the generalizability of our approach, we tested it on the GSM8K dataset, a well-known collection of primary school math problems in English, containing 1,319 test samples. The evaluation results, presented in Table 12, demonstrate that our method yields significant improvements across various metrics, underscoring its strong generalizability.

The high school entrance examination dataset we used covers four subjects—mathematics, biology, chemistry, and physics. Building on the main experiments in the paper, we further present the specific results of our method's Overall metric across these four subjects, as shown in table 13.

Our method achieves optimal results across all subjects and performs consistently well on exam questions spanning various difficulty levels. These results demonstrate that our method generalizes effectively across both different subjects and varying levels of difficulty. The complete set of experiments will be included in the Appendix K.

**Statistical Tests** We conducted three identical experiments on 184 problems and reported the mean and variance of each metric. As shown in Table 14, metrics such as $R_{coverage}$, $F_{frequency}$, Suitability,

Table 12: Results on GSM8K dataset

| | $R_{coverage}$ | $F_{frequency}$ | Suitability | Logic | Inspiration | Reliability | Overall |
|---|---|---|---|---|---|---|---|
| Free-Prompt | 4.23 | 2.32 | 40.00 | 38.24 | 1.23 | 4.68 | 0.95 |
| Stepwise | 3.86 | 3.55 | 41.30 | 50.77 | 2.82 | 4.41 | 2.95 |
| Socratic | 4.14 | 3.91 | 46.15 | 65.22 | 3.27 | 4.00 | 3.59 |
| Bridge-Based | 3.86 | 2.82 | 38.24 | 37.31 | 2.27 | 4.65 | 1.55 |
| Cot-Bridge | 4.32 | 3.77 | 48.28 | 54.35 | 2.55 | 4.73 | 2.59 |
| PELICAN | **4.68** | **4.50** | **63.64** | **74.19** | **3.86** | **4.73** | **4.27** |

Table 13: Overall performance across four subjects.

| | Free-Prompt | Stepwise | Socratic | Bridge-Based | Cot-Bridge | PELICAN |
|---|---|---|---|---|---|---|
| Math | 3.64 | 3.54 | 3.36 | 4.10 | 3.66 | **4.28** |
| Biology | 4.12 | 4.32 | 3.26 | 3.92 | 3.90 | **4.36** |
| Physics | 3.46 | 3.80 | 3.36 | 3.86 | 3.20 | **4.46** |
| Chemistry | 3.00 | 3.44 | 3.15 | 3.84 | 3.47 | **4.13** |

Overall, and Inspiration had small variances, indicating that the observed differences were not due to random fluctuations, but rather demonstrated a certain level of stability and consistency.

### K.1 POWER ANALYSIS

For the Adaptive Tutoring Stage in the main experiment, we performed an ANOVA analysis, with the results summarized in Table 15.

## L PROMPT

```
You are a teacher skilled in diagnosing students' mastery of knowledge points through
↪   problems. Now, a student is unable to solve a problem, and you need to generate a
↪   new problem to assess the student's mastery of a single knowledge point. Based on
↪   the input knowledge point and related content, generate a simple problem to quickly
↪   diagnose the student's mastery level of the knowledge point. The input includes:
- A problem.
- A detailed solution to the problem.
- The knowledge point to be diagnosed.
- The knowledge point chain containing the knowledge point to be diagnosed (used to
↪   infer the meaning of the knowledge point; do not involve subsequent knowledge
↪   points in the diagnosis).
- (Optional) History record: If a history record exists, it indicates that the
↪   previously generated problem or answer may have issues. You need to reflect and
↪   optimize to ensure the correctness of the generated problem. If no history record
↪   exists, disregard this.

Task: Based on the input content, let's think step by step and complete the following
↪   tasks:

1. Based on the problem, solution, and knowledge point chain, determine the level of
↪   assessment for the knowledge point in the problem
↪   (memorization/comprehension/application).
2. Choose an appropriate question type based on the assessment level: (Memorization:
↪   design a fill-in-the-blank question; Comprehension: design a single-choice
↪   question; Application: design a single-choice question, calculation question, or
↪   open-ended question.)
3. Generate a problem that only involves the knowledge point to be diagnosed, with a
↪   difficulty level not exceeding that of the original problem.
4. Provide a concise analysis of the generated problem.
5. Generate the final answer.
```

Table 14: Results of three identical experiments on GAOKAO dataset

| $R_{coverage}$ | $F_{frequency}$ | Suitability | Logic | Inspiration | Reliability | Overall |
|---|---|---|---|---|---|---|
| **55.61**($\pm$4.77) | 64.31($\pm$3.50) | 4.21($\pm$0.002) | 4.37($\pm$0.014) | 4.22(($\pm$0.002)) | 4.31($\pm$0.007) | 4.26($\pm$0.002) |

Table 15: ANOVA Results for Adaptive Tutoring Stage

| Metric | F-value | p-value | Conclusion |
|---|---|---|---|
| Suitability | 3.28 | 0.047 | Significant main effect |
| Logic | 2.97 | 0.031 | Significant main effect |
| Inspiration | 3.14 | 0.015 | Significant main effect |
| Reliability | 3.61 | 0.029 | Significant main effect |
| Overall | 3.59 | 0.042 | Significant main effect |

```
**Output**:
The output must strictly follow the format below:
<step1> Assessment level: xxx
<step2> Question type: xxx
<step3> Problem: xxx
<step4> Analysis: xxx
<step5> Answer: xxx

**Example Output 1**:
<step1> Assessment level: Comprehension
<step2> Question type: Single-choice question
<step3> Problem: In uniformly decelerated linear motion, when the object's velocity
↪  decreases to zero, the characteristic of acceleration is:
A. The direction of acceleration is the same as the direction of velocity
B. The direction of acceleration is opposite to the direction of velocity
C. Acceleration is always positive
D. Acceleration is zero
<step4> Analysis: In uniformly decelerated linear motion, the direction of
↪  acceleration is opposite to the direction of velocity, so the object's velocity
↪  gradually decreases until it reaches zero.
<step5> Answer: B

**Example Output 2**:
<step1> Assessment level: Memorization
<step2> Question type: Fill-in-the-blank question
<step3> Problem: Viruses _ cell structure.
<step4> Analysis: Viruses do not have a cell structure.
<step5> Answer: do not have

# Input
Problem: {problem}
Solution: {analysis}
Knowledge point to be diagnosed: {node_name}
Knowledge point chain: {path}
History (optional): {history}

# Your output:
```

## L.1  STUDENT TYPE EXTRACTION

```
You are a teacher, and you are helping students solve a problem. The problem has been
↪  broken down into multiple independent steps. Each step contains a target question.
↪  You are a student status detection assistant, and you analyze the student's last
↪  round of responses.

Task Description:
```

```
1. Input Content:
   - **Original Problem**: The complete context of the problem.
   - **Original Problem Analysis**: The analysis of the original problem.
   - **Current Step**: The problem to be solved in the current step.
   - **Current Step Analysis**: The analysis of the current step.
   - **Dialogue History**: The historical interaction records between the student and
   ↪  the teacher.

2. Output Content: The output is a JSON-formatted response containing the following
↪  fields:
   - type: Indicates the **response category** corresponding to the student's last
   ↪  response. Choose one from the following categories (output the category number,
   ↪  not the specific content):
     - **Category 1**: The student does not understand the teacher's explanation: The
     ↪  teacher explained knowledge or provided methods/strategies, but the student
     ↪  cannot understand and asks questions about the teacher's explanation.
     - **Category 2**: The student answered incorrectly: The teacher asked a question
     ↪  in the last round, and the student's response contains errors.
     - **Category 3**: The student's mastery of knowledge is poor, and they cannot
     ↪  answer the teacher's last question / The student indicates not knowing a
     ↪  certain knowledge point / The student does not understand the question posed
     ↪  by the teacher.
     - **Category 4**: The student answered correctly: The teacher asked a question in
     ↪  the last round, and the student's response is completely correct.
     - **Category 5**: Other situations not covered above.
   - analysis (considered only for category 4; otherwise, the value is an empty
   ↪  string): First, output the content of the **current step**, then judge based on
   ↪  the student's last response: whether the student's answer sufficiently and
   ↪  accurately answers the question in the **current step**. If it does, output
   ↪  "sufficient"; otherwise, output "insufficient".
   - is_complete (considered only for category 4; otherwise, the value is an empty
   ↪  string): If the student's answer has fully addressed the **current step**,
   ↪  output "1"; otherwise, output "0".

**Output Format**:
The output should be a Markdown code snippet in the following format, including the
↪  markers "```json" and "```".
For example:
{{
    "type": "response category number (1/2/3/4/5)",
    "analysis": "analysis for category 4",
    "is_complete": "0/1"
}}

**Original Problem**: {problem}
**Original Problem Analysis**: {analy}
**Current Step**: {ques}
**Current Step Analysis**: {ans}
**Dialogue History**:
#
{history}
#

Your output:
```

## L.2 STUDENT STATUS UPDATE

```
You are a teacher responsible for updating the student's knowledge state based on
↪  their responses.

**Task**:
```

```
The original problem has been broken down into multiple independent steps, and the
↪  teacher is tutoring the student to solve the **current step**. Please judge and
↪  update the student's hierarchical knowledge state tree based on the student's last
↪  response.

**Question**:
##
{question}
##

**Student's Hierarchical Knowledge State Tree** (use '#' to indicate the hierarchical
↪  structure, and the number following each knowledge point represents the mastery
↪  level: 0=not mastered, 1=mastered):
{state}

**Current Step**:
##
{ques}
##

**Step Analysis**:
##
{ans}
##

**Dialogue History**:
##
{history}
##

**Student's Last Response Type**: {response_type}

**Last Teacher's Purpose**: {last_teacher_purpose}

**Steps**:
1. Analyze the student's last response, response type, and the teacher's purpose to
↪  determine whether related knowledge points need updating:
   - If the student answered correctly, it may indicate that a previously unmastered
   ↪  knowledge point has now been mastered.
   - If the student answered incorrectly, it may indicate that a knowledge point
   ↪  previously marked as mastered has not truly been mastered.
2. Only update the knowledge states you are confident about. Leave uncertain knowledge
↪  points unchanged. If no knowledge points need updating, keep the knowledge state
↪  tree unchanged.

**Output Format**:
The output should be a Markdown code snippet in the following format, including the
↪  markers "```json" and "```".
Please determine whether an update is needed (indicated by yes or no), and then output
↪  the knowledge state tree. Maintain the original structure of the knowledge state
↪  tree, and do not add extra spaces or line breaks. Only modify the values (0 or 1)
↪  of the relevant knowledge points:

{{
    "need_update": "yes/no",
    "state_tree": "Hierarchical Knowledge State Tree"
}}
```

## L.3  TEACHER STRATEGY PREDICT

```
You are a teacher tutoring a student in solving the following problem. The problem has
↪  been broken down into multiple steps, and each step may require multiple rounds of
↪  dialogue to fully resolve. Based on the student's level of understanding, you need
↪  to guide the student step-by-step through the problem-solving process via dialogue.
```

```
**Question**:
##
{question}
##

**Current Decomposed Step**:
##
{ques}
##

**Analysis of the Decomposed Step**:
##
{ans}
##

**Predicted Student Understanding**:
You predict the student's level of understanding of the current step as:
{predict}
- "Can solve independently" indicates that the student has mastered the knowledge of
↪   this step and can be directly asked to answer the question.
- "Lacks relevant understanding" indicates that the student needs more assistance and
↪   requires heuristic tutoring.

**Missing Knowledge Points**:
{missing_knowledge}

**Dialogue History**:
##
{history}
##

**Strategy Used in the Previous Round**:
##
{previous_str}
##

**Purpose of the Previous Round**:
##
{previous_purpose}
##

**Type of Student Response in the Previous Round**:
##
{student_type}
##

**Available Tutoring Strategies**:
a) **Explain a Concept**: Explain a concept or method to the student without directly
↪   providing the answer to the problem.
b) **Suggest a Strategy**: Guide the student to adopt a certain method or strategy to
↪   advance their thinking.
c) **Confirm the Student's Answer**: Acknowledge the student's correct answer and
↪   encourage them to think further about the next step.
d) **Correct the Student's Answer**: Gently point out the student's mistake and guide
↪   them to re-examine the problem for self-correction.
e) **Ask an Open-Ended Question**: Prompt the student to think deeply or complete the
↪   next step of the solution.
f) **Ask a Closed-Ended Question**: Test the student's understanding and memory,
↪   focusing on specific knowledge points.
g) **Simplify the Problem**: Simplify the problem into a more manageable form to help
↪   the student grasp the core idea.
h) **Break Down the Problem**: Divide the problem into smaller sub-problems to help
↪   the student solve it step-by-step.
```

```
i) **Other**.

**Task**:
Based on the above information, choose the tutoring strategy for this round and the
↪  purpose of this round's tutoring.

**Notes**:
1. If the student has multiple missing knowledge points, do not attempt to address all
↪  of them at once. Instead, guide the student step-by-step through multiple rounds
↪  of dialogue.
2. If the predicted understanding level is "Can solve independently," directly ask the
↪  question related to the "Current Decomposed Step."
3. Prioritize guidance over explanation: Encourage the student to think through
↪  various strategies (asking questions, simplifying problems, etc.), helping them
↪  explore the answer through their own reasoning. Provide explanations only when
↪  they encounter significant difficulties.

**Output Format**:
In the `strategy` field, output the strategy code. In the `purpose` field, output the
↪  purpose of this round, such as testing or explaining a specific knowledge
↪  point/concept/skill. Do not set multiple purposes. The output should be a Markdown
↪  code snippet in the following format, including the markers "```json" and "```":
{{
    "strategy": "Your chosen strategy code",
    "purpose": "Your purpose"
}}
```

## L.4 TEACHER FINAL RESPONSE

```
You are a teacher tutoring a student in solving the following problem. The problem has
↪  been broken down into multiple steps, and each step may require several rounds of
↪  dialogue to fully resolve. You need to provide personalized guidance to the
↪  student based on their understanding, helping them gradually solve the 'current
↪  step' through conversation.

**Question**:
##
{question}
##

**Current Step**:
##
{ques}
##

**Step Analysis**:
##
{ans}
##

**Dialogue History**:
##
{history}
##

**Predicted Student Understanding**:
You predict the student's level of understanding of the current step as:
{predict}
- "Can solve independently" means the student has mastered the step and can be
↪  directly asked to answer;
- "Lacks relevant understanding" means the student needs more help and requires
↪  heuristic guidance.

**Knowledge Points Mastered by the Student**:
```

```
{mastered_knowledge}

**Knowledge Points Missing for the Student**:
{missing_knowledge}

**Goal for This Round**:
Advance the student's progress in solving the current step. Please respond to the
↪   student's last message (if there is none, begin your tutoring). You need to adopt
↪   the following strategies and purposes:
- Strategy: {strategy}
- Purpose: {purpose}

**Notes**:
1. **Focus on the Current Step**: If the student asks questions unrelated to the
↪   current step, briefly explain and guide them back to the current step.
2. **Avoid Over-Intervention**: When explaining concepts or providing strategies,
↪   avoid over-intervention. Do not directly give the final answer to the current step.
↪   Avoid excessive hints to help the student think independently.
3. **Single Question at a Time**: Only ask one question per response (if needed). Do
↪   not ask multiple or compound questions.
4. **Personalized Tutoring**: If the student struggles with a certain part, you can
↪   appropriately use phrases like "It seems you have some difficulty
↪   understanding..." to help them identify weak points, but avoid excessive
↪   repetition or rigid expressions.
5. **Avoid Repeating Mastered Content**: Do not repeat teaching content the student
↪   has already mastered, and avoid over-prompting. Focus on what the student has not
↪   yet mastered.

**Output Format**:
The output should be a Markdown code snippet in the following format, including the
↪   markers "```json" and "```":
{{
    "response": "Your response"
}}
```

## L.5   SIMULATE STUDENT RESPONSES

```
You are a student, and you are having trouble solving a problem. A teacher is tutoring
↪   you.
The problem is:
---------------------------------
{question}
---------------------------------

Your knowledge state is:
---------------------------------
{state}
---------------------------------

In the knowledge state, the number of '#' indicates the hierarchical structure of the
↪   knowledge points. The number following each knowledge point represents your
↪   mastery level, where (1) means mastered and (0) means not mastered.

The history record is:
---------------------------------
{history}
---------------------------------

### Task Instructions:
Based on the teacher's last round of tutoring, please assess whether you can
↪   understand the teacher's explanation by considering the following two aspects:
```

```
1. **Changes in History Record**: Review the previous tutoring content and determine
↪  if the teacher's explanation in this round is clearer and more understandable
↪  compared to before. If this round's explanation shows significant improvement or
↪  is better articulated, you can consider yourself able to understand.
2. **Impact of Knowledge State**: Based on your knowledge state, assess whether you
↪  have sufficient foundational knowledge to understand this round's explanation. If
↪  the explanation involves knowledge points you have not mastered or if the teaching
↪  method does not align with your cognitive state, you might find it difficult to
↪  understand.

By combining the history record and your knowledge state, make a judgment and choose
↪  an appropriate response based on the following states:

### Response State Options:
1. **You do not understand a specific knowledge point, process, etc., in the teacher's
↪  explanation**:
   - If you have questions or do not understand certain parts of the teacher's
   ↪  explanation, please raise relevant questions about the parts you do not
   ↪  understand.

2. **You attempt to answer the teacher's question but provide an incorrect answer**:
   - If you attempt to answer the teacher's question but provide an incorrect answer,
   ↪  you can generate an incorrect response that reflects your insufficient
   ↪  understanding of the knowledge point.

3. **Your mastery of the knowledge is poor, and you cannot answer the teacher's
↪  question**:
   - If you feel you lack sufficient knowledge to answer the question, you can
   ↪  directly admit that you cannot answer and express your confusion or difficulty.

4. **You can correctly answer the teacher's question**:
   - If you can accurately answer the teacher's question, you can directly provide the
   ↪  correct answer.

5. **Other**:
   - If your situation does not fit any of the above states, you can choose "Other" and
   ↪  briefly describe your situation.

Output Format:
The output should be a Markdown code snippet in the following format, including the
↪  markers "```json" and "```":
{{
    "can_understand": "yes/no",
    "type": "The response state you chose",
    "response": "Your response"
}}
```