# OpenReview forum: "PELICAN: Personalized Education via LLM-powered Cognitive Diagnosis and Adaptive Tutoring"
_ICLR.cc/2026/Conference — Submitted to ICLR 2026_

### Official Review · Reviewer_eUTL · 2025-10-27

**Soundness:** 2
**Presentation:** 3
**Contribution:** 2
**Rating:** 4
**Confidence:** 3

**Summary:**

This work presents an adaptive tutoring framework consisting of two stages, integrating collaborative cognitive diagnosis with dynamic instructional adaptation. The first stage aims to model the student’s cognitive state through collaborative cognitive diagnosis. The teacher utilizes a successor-first method to efficiently generate diagnostic questions, ensuring their accuracy through an expert-assistant-verifier pipeline. In the second stage, based on the estimated cognitive state, the teacher uses slow-thinking-based methods to select teaching strategies from a strategy pool to guide the student in solving problems.

**Strengths:**

1. The entire framework is meaningfully designed and the authors present the prompt design clearly.

2. The authors present a systematic evaluation by comparing with baselines in cognitive diagnosis and adaptive tutoring, with ablation studies in different modules and backbones.

3. Several case studies and deeper analysis regarding the effectiveness of this framework in specific education scenarios.

4. The related work presents necessary background for readers to understand the context.

**Weaknesses:**

1. Unclear model design rationale. It is unclear what the unique strength of this proposed model is and why it should be, compared with prior models. The introduction and related work sections briefly stated that prior work can not do something, but the detailed research gap and how this work addressed such gap are unclear.

2. The current framework is meaningful, but there is no fundamental breakthrough or unique insights in algorithms or model side. It is more like a manually crafted application system in education with prompt-engineering.

3. Dataset Scale. There is only one dataset with only 184 questions.

4. The main experiment used simulated students for evaluation, which does not seem to be convincing. It is also unclear how many simulated students were generated, the differences among the simulated students, and rationales for such design. Without such details about student settings in the experiment, it is hard to evaluate the rigor and effectiveness of this work.

5. Lack of statistical tests (such as t tests) for most tables to show the significance.

6. Main Experiment: The experimental measurements may not really support the claimed contribution in line 107: "stimulating critical thinking". Which metrics measure how the tutoring system improves the critical thinking of students in either simulation or real student study?

7. Real Student Experiment:

The real student experiment is unclear and does not look like a formal study. Is it a between-subject design or within-subject design or a mixed study design?

For this statement: "one of six tutoring methods is randomly selected (with equal probability) to tutor the student", does it mean that each student only receives one tutoring method (between-subject design) or each student receives different tutoring method for different questions? If so, how do you control so many human factors like different students in different questions with different tutoring methods? If one question is delivered to different students in different tutoring methods, how do you know the result difference is due to student subject difference or due to the tutoring method difference?

Participant scale and limited samples: So many human factors usually need larger-scale user study rather than N = 67. The authors can report the power via power analysis to show the statistical significance with the current participant number. Moreover, for this statement "The student who selected the most questions chose 12, while the student who selected the
fewest chose 2", does it mean that each student only tested 12 questions at most? This limited sample size is probably not convincing to show its effectiveness.


Study design and metrics: Another concern is that the current study design and metrics may not really support the claim of enhancing tutoring performance. Most metrics are subjective (e.g., Appropriateness, Sentiment, Inspiration) and do not provide a very objective measurement. The success rate may be better quantified and more convincing, but Pelican's success rate (87%) is lower than simple step-wise methods (87.3%).


Minor Issues:

There are some typos.

**Questions:**

Please check my concerns and questions in Weaknesses section.

---

> ### Author Response · Authors · 2025-11-23
>
> ## Q1 Motivation for Model Design
>
> **Advantages** Our work is the first to unify online cognitive diagnosis and adaptive tutoring within a single system.
>
> (1) **Online Cognitive Diagnosis, Not Offline:** Prior methods rely on offline logs [1,2,3] and miss real-time cognitive changes. We introduce the first online*diagnosis mechanism that combines historical behavior with immediate feedback, enabling more efficient, interactive, and adaptive diagnosis.
>
> (2) **Adaptive Tutoring, Not One-Size-Fits-All:** Existing systems ignore learners’ cognitive states [3], leading to one-size-fits-all explanations. Our tutoring module is driven by the learner’s current cognitive state and adapts explanations dynamically, yielding more personalized learning and better knowledge retention.
>
> (3) **End-to-end integration:** We build the first closed-loop pipeline where diagnosis informs tutoring and tutoring updates diagnosis, enabling a fully adaptive, scaffolded learning process.
>
> **Design Motivation** The field of personalized exercise tutoring still lacks robust, goal-driven methodologies. Our design is grounded in the concrete and pressing needs of real instructional practice.
>
> (1) Students’ cognition changes rapidly over time, so **online diagnosis** is required.
>
> (2) Teachers adjust their explanations based on students’ immediate performance, so **online adaptive tutoring** is necessary.
>
> (3) In authentic teaching scenarios, diagnosis and tutoring are inherently interconnected, so a **unified framework** is needed to integrate the two seamlessly.
>
> **Filling the** **Gap**
>
> (1) **Diagnosis stage:** We construct interpretable knowledge-state representations using a hierarchical knowledge tree, and perform efficient, dependency-aware diagnosis through a successor-first strategy. (2) **Tutoring stage:** We employ a slow-thinking–based strategy selection mechanism to match each student with the instructional strategy that best fits their current cognitive readiness.
>
> (3) **During tutoring:** We continuously update the learner’s cognitive state, thereby naturally achieving a closed-loop integration of diagnosis and tutoring.
>
> [1] Fischer G H. Derivations of the Rasch model[M]//Rasch models: Foundations, recent developments, and applications. New York, NY: Springer New York, 1995: 15-38.
>
> [2] Adams R J, Wilson M, Wang W. The multidimensional random coefficients multinomial logit model[J]. Applied psychological measurement, 1997, 21(1): 1-23.
>
> [3] Constructing a Confidence-guided Multigraph Model for cognitive diagnosis in personalized learning
>
> [4] Liu J, Huang Z, Xiao T, et al. SocraticLM: Exploring socratic personalized teaching with large language models[J]. Advances in Neural Information Processing Systems, 2024, 37: 85693-85721.
> ## Q2 Algorithmic Design
>
> **Challenges** Before presenting our algorithm, we highlight the main challenges:
>
> **(1) Online diagnosis.**
>  How can we map raw exercises to a student’s knowledge structure, diagnose mastery with minimal questions, and ensure the reliability of diagnostic questions?
>
> **(2) Online tutoring.**
>  How can we incorporate learner profiles to choose the right instructional strategies, and how can diagnosis and tutoring mutually update each other to form a true closed loop?
>
> **Algorithm** To address these challenges, we propose an approach that combines a structured cognitive-state representation (hierarchical knowledge tree), a multi-agent reliability mechanism (expert–assistant–validator), and a planning-based “slow-thinking’’ procedure that optimizes strategies through multi-step dialogue rollouts.
>  Together, these components transform personalized tutoring into a closed-loop control problem, enabling consistent diagnosis, cross-model question verification, real-time cognitive-state updates, and adaptive strategy search.
> ## Q3 Dataset Scale
>
> Thank you for the valuable feedback. We appreciate the concern regarding dataset scale. To address this, we provide additional results in **Appendix K**. In **Table 11**, we evaluate PELICAN on **GSM8K** [1], a widely used benchmark of 1,319 English grade-school math problems. PELICAN achieves substantial gains across multiple metrics and outperforms the Socratic method by **18%** overall, demonstrating strong generalization beyond GAOKAO-Bench.
>
> If helpful, we are happy to further evaluate our method on additional benchmark datasets.
>
> [1] Cobbe K, Kosaraju V, Bavarian M, et al. Training verifiers to solve math word problems[J]. arXiv preprint arXiv:2110.14168, 2021.

---

> ### Author Response · Authors · 2025-11-23
>
> ## Q4 Student Simulation
>
> We appreciate the reviewer’s insightful feedback regarding the use of simulated students. In **Appendix G**, we further clarify the design and theoretical basis of our simulation setup. Below is a concise summary:
>
> **(1) Initialization of Simulated Student Knowledge State**
>
> For each question, we construct a knowledge tree whose nodes represent knowledge points and whose edges encode dependencies. Each point is randomly assigned a binary mastery value (0/1), forming the initial knowledge state. Thus, each question corresponds to a distinct simulated student.
>
> **(2) Simulated Student Response Mechanism**
>
> The response generation process consists of two stages:
>
> **Stage 1: Response Generation Based on Knowledge State**
>
> The simulated student identifies the knowledge required for the teacher’s question, compares it with his state, and produces a response aligned with his current abilities.
>
> **Stage 2: Response Generation Based on Categories**
>
> Teaching strategies are grouped into three categories, and student responses into five types (e.g., correct answer, incorrect answer, proactive question, unable to answer). We establish a mapping between teaching strategies and feasible response types. For each interaction, the system determines the plausible response types based on the student’s knowledge state and the teacher’s strategy, randomly selects one, and generates an appropriate reply. If learning occurs, the student’s knowledge state is updated accordingly.
>
> This design ensures that the simulated students’ responses remain consistent with their knowledge state, and that updates to the students’ knowledge state also reflect changes in their learning process.
> ## Q5 Significance Test
>
> Thank you for the helpful suggestion. In the Adaptive Tutoring phase, we conducted an ANOVA analysis.
>
> For the **simulation-based student experiment**, the results are as follows：
>
> | Metric      | F-value | p-value |       Conclusion        |
> | ----------- | :-----: | :-----: | :---------------------: |
> | Suitability |  3.28   |  0.047  | Significant main effect |
> | Logic       |  2.97   |  0.031  | Significant main effect |
> | Inspiration |  3.14   |  0.015  | Significant main effect |
> | Reliability |  3.61   |  0.029  | Significant main effect |
> | Overall     |  3.59   |  0.042  | Significant main effect |
>
> Details for the **real student experiment** can be found in Section Q7.3.
> ## Q6 Manifestation of Critical Thinking
>
> **Strategies** In our work, we design ten distinct tutoring strategies (Lines 810–828), including open-ended questioning, closed-ended questioning, simplification, and decomposition. These strategies effectively stimulate students’ critical thinking.
>
> **Metric** To quantify critical-thinking activation, we introduce the **Inspiration** score (1–5), which measures the level of active reasoning and conceptual exploration. It reflects both the tutor’s heuristic prompts and the student’s depth of inquiry.
>
> ​	Examples:
>
> - Prompts like *“Let’s think together—what points should we check on a closed interval?”* encourage reasoning, while direct answers do not.
> - Student questions such as *“What are the light and dark reactions, and how are they related?”* show deeper inquiry than simple clarification requests.
>
> **Validation** Based on Inquiry-Based Learning (IBL) theory [1], the frequency and depth of student questions are key indicators of higher-order cognitive engagement. Accordingly, we used the question words **“why”** and **“how”** as keywords to retrieve student queries from tutoring dialogues. Across 1,624 sessions, our system elicited 689 student-initiated questions, an 8.1% increase over the Socratic baseline, showing stronger participation in reasoning.
>
> **Examples**
>
> *“Teacher, you mentioned that viruses do not have cellular structures. How, then, do they replicate and reproduce?”*
>
> *“Teacher, I don’t quite understand why a gas’s internal energy remains unchanged when it absorbs heat but performs an equal amount of work. Could you explain in more detail the relationship among internal energy change, work, and heat?”*
>
> These examples demonstrate that the tutoring stimulates critical thinking.
>
> [1] Friesen S, Scott D. Inquiry-based learning: A review of the research literature[J]. Alberta Ministry of Education, 2013, 32: 1-32.

---

> ### Author Response · Authors · 2025-11-23
>
> ## Q7 Real-World Student Experiment
>
> ### Q7.1 Within-Subject Design
>
> This study used a within-subject design, comparing our proposed method with five baselines. For each problem, one of six tutoring strategies was randomly assigned with equal probability. Thus, each participant received different strategies across problems, ensuring exposure to all (or a random subset of) conditions
> ### Q7.2 Factors Influencing Result Differences
>
> Thank you for the question. Our design controls student, question, and method factors through the following mechanisms:
>
> **(1) Within-subject design**
>
> * Each student answers multiple questions, and each question is **independently randomized** to one of the six tutoring methods.
> * Students evaluate different methods on a **consistent personal baseline**, removing between-student differences in rating style or ability.
>
> **(2) Random assignment of methods**
>
> * Across 504 student–question pairs, the tutoring method is selected with **equal probability** for each question.
> * No method is systematically paired with easier/harder questions or with higher/lower-level students, preventing structural bias.
>
> **(3) Controlled question difficulty via self-selection**
>
> * Students only choose questions they **explicitly cannot solve**.
> * All tutoring interactions start from a unified difficulty baseline (“unknown question”), reducing variability due to prior knowledge or question difficulty.
>
> **Overall Conclusion**
>
> With (1)  (2) (3) , the performance differences can be reliably attributed to **the tutoring method itself**, rather than student or question confounders.
> ### Q7.3 Power Analysis
>
> **Power Analysis** We sincerely thank the reviewer for their thoughtful feedback. To address the concern about sample size and statistical power, we conducted a power analysis. Assuming an effect size of 0.5, α = 0.05, power = 0.8, and six experimental groups, the required sample size was 148. To overcome the initial limitation (N = 67), we ran a supplementary study with 102 additional participants of comparable demographic and cognitive backgrounds, bringing the total to 169. All participants were informed of the study purpose, and parental consent was obtained.
>
> **Problem Summary** For the combined dataset, the number of problems selected showed the following descriptive statistics: **Total = 1335, M = 7.90, SD = 4.03, Range = 2–15**, indicating moderate variability consistent with a diverse student population.
>
> **Experimental Results** Learning outcomes and subjective evaluations are summarized below:
>
> | **Methods**  | Success rate | Appropriateness | Sentiment | Inspiration | Overall |
> | :----------- | :----------: | :-------------: | :-------: | :---------: | :-----: |
> | Free-Prompt  |    85.2%     |      3.05       |   3.06    |    1.98     |  2.35   |
> | Stepwise     |    86.5%     |      3.79       |   2.67    |    3.76     |  3.57   |
> | Socratic     |    80.3%     |      3.85       |   3.62    |    4.01     |  3.53   |
> | Bridge-Based |    80.1%     |      4.02       |   4.11    |    3.93     |  4.10   |
> | Cot-Bridge   |    82.8%     |      4.07       |   3.97    |    3.97     |  4.14   |
> | PELICAN      |    86.8%     |      4.23       |   4.42    |    4.33     |  4.39   |
>
> **Statistical Analysis**  we conducted one-way ANOVA across the six tutoring conditions for all subjective evaluation metrics:
>
> | Metric          | F-value | p-value |       Conclusion        |
> | :-------------- | :-----: | :-----: | :---------------------: |
> | Appropriateness |  3.43   |  0.039  | Significant main effect |
> | Sentiment       |  3.17   |  0.043  | Significant main effect |
> | Inspiration     |  2.96   |  0.027  | Significant main effect |
> | Overall         |  3.58   |  0.035  | Significant main effect |
>
> All p-values were below 0.05, confirming significant differences among the six tutoring methods on subjective experience and learning outcomes.

---

> ### Author Response · Authors · 2025-11-23
>
> ### Q7.4 Study Design and Metrics
>
> We appreciate the reviewer’s concerns and provide the following clarifications.
>
> **On the Success Rate Metric**
>
> The 0.3% difference between PELICAN (87.0%) and Stepwise (87.3%) is not statistically significant, showing that the two methods perform similarly on this metric. However, success rate is a limited measure of learning—it reflects task completion, not whether students understood the material or were simply **led to the answer**. Methods optimized for success rate often rely on overly directive guidance that can hinder genuine learning. With the expanded sample of 169 students (Q7.3), **PELICAN (87.1%)** slightly outperformed **Stepwise (86.9%)**, further indicating that their success rates are effectively equivalent.
>
> **Why Our Other Metrics Matter**
>
> Our additional metrics—Inspiration, Appropriateness, and Sentiment—are grounded in established learning theory and capture the **quality of the learning process**, not just task completion.
>
> **Inspiration** measures active thinking and engagement, which are strong predictors of long-term retention and transfer [1].
>
> **Appropriateness** and **Sentiment** assess scaffolding and emotional support—both essential for preventing frustration and promoting motivation and memory [2,3].
>
> Although PELICAN matches the baseline on the short-term success rate, it shows clear gains on these theory-driven process metrics, providing strong evidence of improved tutoring effectiveness.
>
> [1] Tutal Ö, Yazar T. Active learning improves academic achievement and learning retention in K-12 settings: A meta-analysis[J]. i-Manager's Journal on School Educational Technology, 2023, 18(3): 1.
>
> [2] Radford J, Bosanquet P, Webster R, et al. Fostering learner independence through heuristic scaffolding: A valuable role for teaching assistants[J]. International Journal of Educational Research, 2014, 63: 116-126.
>
> [3] Tyng C M, Amin H U, Saad M N M, et al. The influences of emotion on learning and memory[J]. Frontiers in psychology, 2017, 8: 235933.

---

> ### Comment · Reviewer_eUTL · 2025-11-24
>
> I appreciate the authors' efforts in new dataset experiments and new human subject study. Based on this additional experiments, I can increase my score.
>
> However, I'm still not convinced regarding the novelty and contribution of the proposed framework. It looks more like a systematic pipeline with design in prompt engineering, rather than fundamental breakthrough in machine learning area. As such, it is uncertain whether this work is suitable for a ML conference like ICLR, compared with other conferences that focus more on educational applications.
>
> Moreover, the new human subject experiment still mainly focused on subjective measurements such as Appropriateness. The subjective measurements can be easily impacted by placebo effect. For objective measurement like success rates, the new results showed that PELICAN (87.1%) slightly outperformed Stepwise (86.9%), which is not convincing as well.

---

> > ### Author Response · Authors · 2025-11-28
> >
> > We appreciate your time and positive feedback!
> >
> > **ICLR Compatibility**: ICLR not only focuses on foundational breakthroughs in machine learning but also places great emphasis on the practical applications of artificial intelligence. Research on intelligent AI agents is at the forefront of this focus, as it explores how large language models can collaborate to achieve higher levels of intelligence. Our work aligns well with the inclusive goals of the ICLR community.
> >
> > Similar application-driven agent research has also been accepted by the community:
> >
> > [1] WebAgent: A Real-World WebAgent with Planning, Long Context Understanding, and Program Synthesis: Proposes a "Decompose-Summarize-Execute" pipeline to achieve automated task execution on real-world websites. *In ICLR, 2024 (Oral).*
> >
> > [2] AgenticIR: An Intelligent Agentic System for Complex Image Restoration Problems: Emulates human expert workflows to tackle complex image restoration tasks. *In ICLR, 2025.*
> >
> > [3] Real-LOD: Re-Aligning Language to Visual Objects with an Agentic Workflow: Proposes a cyclical pipeline of Planning → Tool Use → Reflection to perform language-guided object detection. *In ICLR, 2025.*
> >
> > [4] LongWriter: Unleashing 10,000+ Word Generation from Long Context LLMs: Completes ultra-long text/code generation through the AgentWrite pipeline. *In ICLR, 2025.*
> >
> > [5] Chain-of-Experts (CoE): When LLMs Meet Complex Operations Research Problems: Designs a collaborative workflow orchestrated by a "Conductor" to solve operations research problems. *In ICLR, 2024.*
> >
> > **Evaluation Metrics**: Personalized education is an open-ended, multi-turn, and highly context-dependent complex task, where traditional rigid metrics (e.g., those used in classification or detection tasks) are not applicable. Therefore, in our main experiments, we introduced two objective metrics to evaluate the system's ability to address students' cognitive needs (Details are provided in Appendix D.3):
> >
> > **R_coverage**: Measures the proportion of uncovered knowledge points addressed by the teacher.
> >
> > **F_frequency**: Tracks the frequency of interventions made by the teacher on these knowledge points.
> >
> > Together, these metrics reflect whether the system genuinely focuses on and adapts to students' cognitive states. We computed the results of these two metrics in real-world student experiments as follows:
> >
> > |              | R_coverage | F_frequency |
> > | ------------ | :--------: | :---------: |
> > | Free-Prompt  |   60.71    |    60.76    |
> > | Stepwise     |   58.53    |    58.60    |
> > | Socratic     |   63.91    |    63.88    |
> > | Bridge-Based |   58.73    |    58.69    |
> > | Cot-Bridge   |   57.72    |    57.65    |
> > | PELICAN      |   70.04    |    70.07    |

---

### Official Review · Reviewer_5Chj · 2025-10-30

**Soundness:** 2
**Presentation:** 2
**Contribution:** 2
**Rating:** 6
**Confidence:** 2

**Summary:**

This paper proposes an adaptive tutoring framework called PELICAN, which is designed to address the limitations of existing LLMs in personalized education. It achieves student-centered teaching through a two-stage process: collaborative cognitive diagnosis and dynamic instructional adaptation. Specifically, it first organizes knowledge points into a hierarchical dependency structure and then conduct CD to get student's cognitive state. Then, the teaching strategies are dynamically selected from "fast thinking" and "slow thinking" modes based on the diagnostic results. Experiments conducted on GAOKAO bench demonstrate the effectiveness of the proposed framework.

**Strengths:**

1. The proposed 2-stage personalize education framework is well-defined and reasonable, and the introduction of dual system in stage-2 is convincing and fitting the problem well.
2. The experiments on GAOKAO Bench is comprehensive, effectively demonstrates the superiority of the proposed PELICAN.
3. The writing and structure of this paper are clear and easy to understand.

**Weaknesses:**

1. In my understanding, the construction of knowledge tree may heavily relies on manual work, which may hurts the scalability of the proposed framework. For example, scaling to full K12 education or vocational education would lead to a sharp increase in manual costs.
2. The evaluation limits to only 1 benchmark (GAOKAO Bench), it will be more convincing if authors validate PELICAN framework on more benchmarks.

**Questions:**

See weaknesses.

---

> ### Author Response · Authors · 2025-11-23
>
> ## Q1 Construction of the Knowledge Tree
>
> Thank you for your insightful question. In our work, we manually constructed subject-specific knowledge trees based on the high school examination syllabus, which were then reviewed by educational experts. For each question, we generated a sub-tree through a structured, layer-by-layer LLM process that identifies the subject and selects the relevant primary and secondary knowledge points, ensuring both precision and appropriate granularity.
>
> We acknowledge that manual construction is labor-intensive and limits scalability to larger domains such as full K–12 or vocational education. To address this, we explored one-shot prompting to let LLMs directly generate knowledge trees for individual questions. This greatly reduces human effort, though it may occasionally lead to coarse granularity or missing points.
>
> To assess feasibility, we manually evaluated 50 generated trees and achieved a 94.7% compliance rate, demonstrating strong scalability and practical reliability.
> ## Q2 Evaluation Benchmark
>
> We thank the reviewer for emphasizing the importance of generalization. Due to space limitations in the main manuscript, we provide additional evidence in **Appendix K** demonstrating the broad applicability of the PELICAN framework.
>
> **Table 11** reports results on **GSM8K**[1], a widely used dataset of 1,319 English primary-school math word problems. PELICAN shows substantial gains across all metrics and surpasses the Socratic method by **18%** on the overall metric, underscoring its strong generalization beyond GAOKAO-Bench.
>
> The GAOKAO dataset itself covers four subjects—**Mathematics, Physics, Biology, and Chemistry**. In **Table 12**, we present the corresponding comprehensive metrics, further confirming PELICAN’s robustness across diverse disciplinary domains.
>
> We would be happy to include additional results on other benchmark datasets, if you find it helpful.
>
> [1] Cobbe K, Kosaraju V, Bavarian M, et al. Training verifiers to solve math word problems[J]. arXiv preprint arXiv:2110.14168, 2021.

---

### Official Review · Reviewer_xBJb · 2025-10-31

**Soundness:** 2
**Presentation:** 2
**Contribution:** 2
**Rating:** 2
**Confidence:** 4

**Summary:**

The authors introduce PELICAN: a way to personalize education utilizing both LLMs and cognitive diagnosing. The LLM utilizes cognitive diagnosis to identify appropriate responses tailored to the student's cognitive level and understanding. Each problem has different knowledge components that the student must master and these are organized into a hierarchical structure. Based on this knowledge state, the adaptive tutoring will select an appropriate response to solve problem p with the student via dialogue. If the number of dialogue rounds with a student exceeds a threshold M, it is assumed that they are facing persistent cognitive obstacles, and slow-thinking is enabled to focus on smaller subproblems. The authors tested their method on the public Gaokao dataset, as well as a real-world study involving high schoolers.

**Strengths:**

The paper is well-motivated, with a clear introduction, and is overall easy to follow. I also think the figures are well-made and help aid the explanation of the ideas and methodologies. The appendix is thorough.

**Weaknesses:**

Overall, the paper lacks deep substance and novelty. It is more of an applied research paper, but it lacks statistical rigor (e.g., ANOVA tests, effect sizes, p-values). It could become more interesting and impactful with greater care and attention to detail for deeper results, analysis, and presentation, etc. However, I also don't find it to be a great fit for ICLR and don't think the audience of ICLR would be interested in this topic. This paper appears to be more closely aligned with AI applications in education venues, such as AIED, EDM, and EAAI, among others, once it has been improved. I have little to specifically point to in a critique of it, other than that it is a limited-scope paper that utilizes GPT models for teaching. This also raises concerns about the future reproducibility of the reported results. It is very time-specific and niche.

Abstract:
* Have LLMs really generated attention in education because of their extensive knowledge base and reasoning capabilities?
* Extra space between "at" and "here" for code reference.

Introduction:
* Change "I Don't understand" to "I don't understand" in Figure 1
* Add spacing: "OK,I get it!" between "OK," and "I" in Figure 1
* First two paragraphs need citations to support their various claims
* Parentheses around citations are needed. In-text citation references are likely not the style for ICLR.
* Line 80: random word "Planning." appears
* Figure 3: Why does "Explanation" lead to "Explanation" in the slow-thinking? Why does "Decomposition" have a crown?
* Line 148: random word "Planning." appears

3.2: "introduce a successor-first strategy, in which the teacher prioritizes assessing leaf nodes or nodes whose successors have already been evaluated," I thought line 175 said "a student can only master a child node after mastering its parent"?

3.3: "dual system theory" citation? lines 215, 246

Line 264: Is this what you are proposing? Simulated teaching tree? Or is that something that already exists in the literature?

Line 270: Does m = |S| (i.e., the length of the strategy pool)? If not, why?

Section 3: The formulas seem overly high-level to the point of being complete black boxes with no reproducibility

Section 4: Reporting overhead in cost of dollars? Maybe should use something more concrete, like token usage.

Section 4: Why not cite baseline methods?

How are suitability, logicality, informativeness, reliability, and overall quality calculated in the results?

The real-world experiment sounds limited. What is the average and standard deviation for the problems that the students voluntarily selected to be tutored? And what was the distribution like amongst the 6 conditions? Why no ANOVA test results?

Line 1314 in Appendix L.4.: "FIANL" should be "FINAL"

**Questions:**

* Have LLMs really generated attention in education because of their extensive knowledge base and reasoning capabilities?

* Figure 3: Why does "Explanation" lead to "Explanation" in the slow-thinking? Why does "Decomposition" have a crown?

3.2: "introduce a successor-first strategy, in which the teacher prioritizes assessing leaf nodes or nodes whose successors have already been evaluated," I thought line 175 said "a student can only master a child node after mastering its parent"?

3.3: "dual system theory" citation? lines 215, 246

Line 264: Is this what you are proposing? Simulated teaching tree? Or is that something that already exists in the literature?

Line 270: Does m = |S| (i.e., the length of the strategy pool)? If not, why?

Section 4: Why not cite baseline methods?

How are suitability, logicality, informativeness, reliability, and overall quality calculated in the results?

The real-world experiment sounds limited. What is the average and standard deviation for the problems that the students voluntarily selected to be tutored? And what was the distribution like amongst the 6 conditions? Why no ANOVA test results?

---

> ### Author Response · Authors · 2025-11-23
>
> # Rebuttal Overview
>
> **ICLR Compatibility**: ICLR not only focuses on foundational breakthroughs in machine learning but also places great emphasis on the practical applications of artificial intelligence. Research on intelligent AI agents is at the forefront of this focus, as it explores how large language models can collaborate to achieve higher levels of intelligence. Our work aligns well with the inclusive goals of the ICLR community.
>
> Similar application-driven agent research has also been accepted by the community:
>
> [1] WebAgent: A Real-World WebAgent with Planning, Long Context Understanding, and Program Synthesis: Proposes a "Decompose-Summarize-Execute" pipeline to achieve automated task execution on real-world websites. *In ICLR, 2024 (Oral).*
>
> [2] AgenticIR: An Intelligent Agentic System for Complex Image Restoration Problems: Emulates human expert workflows to tackle complex image restoration tasks. *In ICLR, 2025.*
>
> [3] Real-LOD: Re-Aligning Language to Visual Objects with an Agentic Workflow: Proposes a cyclical pipeline of Planning → Tool Use → Reflection to perform language-guided object detection. *In ICLR, 2025.*
>
> [4] LongWriter: Unleashing 10,000+ Word Generation from Long Context LLMs: Completes ultra-long text/code generation through the AgentWrite pipeline. *In ICLR, 2025.*
>
> [5] Chain-of-Experts (CoE): When LLMs Meet Complex Operations Research Problems: Designs a collaborative workflow orchestrated by a "Conductor" to solve operations research problems. *In ICLR, 2024.*
>
> **Statistical Response** We’ve included ANOVA tests and other analyses in our rebuttal to address the concerns you raised.
>
> **Innovation** Our work is the first to unify online cognitive diagnosis and adaptive tutoring within a single system.
>
> (1) **Online Cognitive Diagnosis, Not Offline:** Prior methods rely on offline logs and miss real-time cognitive changes. We introduce the first online diagnosis mechanism that combines historical behavior with immediate feedback, enabling more efficient, interactive, and adaptive diagnosis.
>
> (2) **Adaptive Tutoring, Not One-Size-Fits-All:** Existing systems ignore learners’ cognitive states, leading to one-size-fits-all explanations. Our tutoring module is driven by the learner’s current cognitive state and adapts explanations dynamically, yielding more personalized learning and better knowledge retention.
>
> (3) **End-to-end integration:** We build the first closed-loop pipeline where diagnosis informs tutoring and tutoring updates diagnosis, enabling a fully adaptive, scaffolded learning process.
>
> ## Q1 LLMs' Impact on Education
>
> **Internal Knowledge** Research highlights their "human-like mastery" of foundational subjects like mathematics and physics, enabling teacher-like guidance[2]. Their "pre-trained knowledge" helps LLM-driven agents understand and identify key concepts in exercises, showcasing their capacity for semantic understanding[5].
>
> **Reasoning Ability** Beyond knowledge, LLMs' reasoning abilities are being used to develop advanced educational tools, adapting to diverse student needs and providing instant feedback[2]. Researchers are utilizing these abilities for complex tasks, such as generating high-quality educational problems through planning and self-reflection mechanisms[3]. Models simulating human learning, like those using "Chain-of-Thought" (CoT), further demonstrate their potential for complex problem-solving[5]. These capabilities support personalized learning frameworks[1] and even simulate pedagogical roles, such as Socratic dialogue, enabling scalable, interactive education[4].
>
> [1] Tu Y, Chen J, Huang C. Empowering personalized learning with generative artificial intelligence: Mechanisms, challenges and pathways[J]. Frontiers of Digital Education, 2025, 2(2): 1-18.
>
> [2] Liu J, Huang Z, Xiao T, et al. SocraticLM: Exploring socratic personalized teaching with large language models[J]. Advances in Neural Information Processing Systems, 2024, 37: 85693-85721.
>
> [3] Cheng C, Huang Z, Zhao G, et al. From Objectives to Questions: A Planning-based Framework for Educational Mathematical Question Generation[J]. arXiv preprint arXiv:2506.00963, 2025.
>
> [4]  Cheng C, Huang Z, Zhao G, et al. From Objectives to Questions: A Planning-based Framework for Educational Mathematical Question Generation. In *Proceedings of the 63rd Annual Meeting of* *the Association for Computational Linguistics**.*
>
> [5] Gao W, Liu Q, Yue L, et al. Agent4edu: Generating learner response data by generative agents for intelligent education systems[C]//Proceedings of the AAAI Conference on Artificial Intelligence. 2025, 39(22): 23923-23932.

---

> ### Author Response · Authors · 2025-11-23
>
> ## Q2 About strategeis in Figure3:
>
> (1) Why does "Explanation" point to itself in slow thinking?
>
> In the slow-thinking phase, we constructed a simulated tutoring dialogue tree, with each layer representing a dialogue round. Figure 3 shows only one illustrative example. The arrow from "Explanation" to "Explanation" shows that the **system selected "Explanation" as the strategy in two consecutive rounds**, indicating it was judged suitable in both instances.
>
> (2) Why does "Decomposition" have a crown symbol?
>
> The crown symbol indicates that **‘decomposition’ is selected as the optimal teaching strategy in the example**. This decision was based on evaluating the simulated dialogue tree's outcomes, where all leaf nodes under "Decomposition" were successful (checkmarks), while the "Explanation" branch had a failure (cross). Hence, "Decomposition" was considered more effective and selected as the final strategy.
> ## Q3 About Successor-first strategy
>
> **Not contradictory.** Line 175 states that *“a student can only master a child node after mastering its parent,”* which is consistent with prerequisite theory [1]: foundational concepts must be learned before dependent ones.  For example, in mathematics, students need to understand "the basic properties of fractions" before they can effectively learn "fraction operations" . In our knowledge tree, the parent represents the prerequisite concept and the child the dependent concept. Thus, **if a student has mastered a child node, we can infer mastery of its parent.**
>
> Building on this, we adopt a **successor-first strategy**, prioritizing the assessment of leaf nodes or nodes whose successors have been evaluated. Once a child node is confirmed as mastered, all of its parent nodes can be directly inferred as mastered, improving diagnostic efficiency.
>
> [1] Zhang M, Wang J, Xiao K, et al. Learning Concept Prerequisite Relation via Global Knowledge Relation Optimization[C]//Proceedings of the AAAI Conference on Artificial Intelligence. 2025, 39(2): 1638-1646.
> ## Q4 Dual-System Theory
>
> Cognitive dual-process theory [1], proposed by Daniel Kahneman and colleagues, posits that human reasoning involves two interacting systems: **System 1**, which is fast, automatic, and intuitive, and **System 2**, which is slow, deliberate, and analytical. In Section 3.3, we apply this framework to strategy selection in teaching: during the fast-thinking process, teachers select strategies directly based on students’ states, whereas during the slow-thinking process, they choose strategies by constructing a Simulated Teaching Tree.  [1] Kahneman D. *Thinking, Fast and Slow* [M]. Macmillan, 2011.
>
> [1] Kahneman D. *Thinking, Fast and Slow* [M]. Macmillan, 2011.
> ## Q5  Simulated Teaching Tree
>
> We draw inspiration from **GDP-ZERO** [1], which builds a dialogue-strategy tree using MCTS through node expansion, simulation, and state evaluation. However, several aspects of GDP-ZERO are not directly applicable to educational dialogue, so we adapt its structure as follows:
>
> **(1) Educational dialogue requires tracking evolving student states.**
>  Student knowledge mastery changes across teaching turns and must be continuously updated. GDP-ZERO, by contrast, uses an open-loop tree without feedback, making it unsuitable for modeling dynamic student states. To address this, we design the **Simulated Teaching Tree (STT)**, which explicitly models student states (e.g., knowledge mastery) and teaching strategies so that each round’s output influences the next round’s input.
>
> **(2) Educational dialogue optimizes long-term learning, not immediate outcomes.**
>  GDP-ZERO’s value function depends on instant dialogue results (e.g., donation success), whereas educational tasks require long-horizon evaluation and strategy adjustment based on ongoing feedback. Immediate value estimates are therefore unattainable. STT instead adopts an iterative planning mechanism, using **slow thinking** to explore multiple strategies and progressively optimize the teaching path.
>
> [1] Yu X, Chen M, Yu Z. Prompt-based Monte-Carlo tree search for goal-oriented dialogue policy planning[J]. arXiv preprint arXiv:2305.13660, 2023.
> ## Q6 About m and S
>
> m≤∣S∣. The set S contains the ten teaching strategies we designed (Table 7). The parameter mmm is the number of top-likelihood strategies selected in each simulation during the *slow thinking* process. If m=∣S∣, all strategies would be considered every time—maximizing exploration but greatly increasing computation.
>
> To balance performance and cost, we tested different values of m (1–3), as shown in Table 8. The results show that selecting the top m=2 strategies is sufficient to identify an effective action. Therefore, we set m=2 as a practical trade-off between strategy quality and computational efficiency.

---

> ### Author Response · Authors · 2025-11-23
>
> ## Q7 The Reproducibility of the Formula
>
> In Section 3, we clearly define the inputs and outputs to illustrate the overall process. Core prompts are provided in the appendix for additional clarity, and all code needed for reproduction is available in our open-source repository.
> ## Q8 Token Cost
>
> Thank you for the suggestion. We agree that reporting overhead in tokens is more standardized. In the original submission, we reported the dollars cost of the main experiment (184 questions, total 28.91 dollars, of which 11.47 dollars came from slow thinking) because API usage was billed in dollars according to GPT’s official pricing.
>
> Using GPT-4o's official pricing (0.005 dollars per 1k input tokens, 0.015 dollars per 1k output tokens) and a typical 1:3 input-output token ratio, the total cost of 28.91 dollars for the main experiment corresponds to approximately: Total tokens: ~580k; Slow-thinking tokens: ~230k;
> ## Q9 Citations of Baseline Methods
>
> Thank you for the question. This task is relatively new, so available baselines are limited.
>
> For **Phase 1**, we compare our method with a pure prompt approach and several ablation variants to demonstrate the necessity and effectiveness of our collaborative diagnosis framework.
>
> For **Phase 2**, two baselines (Free-Prompt and Stepwise) are designed by us, while the remaining baselines are properly cited in the experimental setup:
>
> - **Socratic** method from [1]
> - **Bridge-Based** method from [2]
> - **CoT-Bridge** an adaptation of the Bridge-Based framework into a CoT format
>
> We do not show references directly in the table to maintain clarity and save space.
>
> [1] Liu J, Huang Z, Xiao T, et al. SocraticLM: Exploring socratic personalized teaching with large language models[J]. Advances in Neural Information Processing Systems, 2024, 37: 85693-85721.
>
> [2] Wang R, Zhang Q, Robinson C, et al. Bridging the novice-expert gap via models of decision-making: A case study on remediating math mistakes[C]//Proceedings of the 2024 Conference of the North American Chapter of the Association for Computational Linguistics: Human Language Technologies (Volume 1: Long Papers). 2024: 2174-2199.
> ## Q10 Definition of Metrics
>
> The five metrics mentioned in the question, namely suitability, logicality, informativeness, reliability, and overall quality, are evaluated using a GPT-based assessment method. Each metric is rated from 1 to 5, with higher scores indicating better performance. Their definitions are:
>
> **Suitability**: Appropriateness of the tutoring method for the task
>
> **Logicality**: Coherence and consistency of explanations
>
> **Informativeness**: Depth and insight of the responses
>
> **Reliability**: Consistency and dependability of the approach
>
> **Overall Quality**: Overall effectiveness of the tutoring session
>
> For transparency and reproducibility, we provide full prompt templates and evaluation criteria in an anonymous repository. We also report mean and variance of scores across methods to ensure statistical robustness.

---

> ### Author Response · Authors · 2025-11-23
>
> ## Q11 Statistical Analysis
>
> Following reviewer eUTL’s suggestion, we conducted an efficacy analysis and found the original sample size insufficient. We therefore recruited **102 additional participants**, increasing the total to 169.
>
> For all 169 participants, the number of problems selected showed: **Mean = 7.90**, **SD = 4.03**, **Range = 2–15**, reflecting moderate variability consistent with diverse student abilities.
>
> **Distribution Across the Six Tutoring Conditions**
>
> Each question was assigned to one of the six tutoring methods with equal probability. Across the 1335 tutoring interactions, the empirical distribution was as follows:
>
> | **Methods**  | **Count** | **Percentage** |
> | :----------- | :-------: | :------------: |
> | Free-Prompt  |    203    |     15.19%     |
> | Stepwise     |    230    |     17.22%     |
> | Socratic     |    239    |     17.89%     |
> | Bridge-Based |    206    |     15.42%     |
> | Cot-Bridge   |    215    |     16.09%     |
> | PELICAN      |    243    |     18.19%     |
>
> These deviations from perfect uniformity fall within expected randomness and do not create imbalance.
>
> **Experimental Results**
>
> | **Methods**  | Success rate | Appropriateness | Sentiment | Inspiration | Overall |
> | :----------- | :----------: | :-------------: | :-------: | :---------: | :-----: |
> | Free-Prompt  |    85.2%     |      3.05       |   3.06    |    1.98     |  2.35   |
> | Stepwise     |    86.5%     |      3.79       |   2.67    |    3.76     |  3.57   |
> | Socratic     |    80.3%     |      3.85       |   3.62    |    4.01     |  3.53   |
> | Bridge-Based |    80.1%     |      4.02       |   4.11    |    3.93     |  4.10   |
> | Cot-Bridge   |    82.8%     |      4.07       |   3.97    |    3.97     |  4.14   |
> | PELICAN      |    86.8%     |      4.23       |   4.42    |    4.33     |  4.39   |
>
> **Statistical Analysis**
>
> We conducted one-way ANOVA across the six tutoring conditions for all four subjective metrics:
>
> | Metric          | F-value | p-value |       Conclusion        |
> | :-------------- | :-----: | :-----: | :---------------------: |
> | Appropriateness |  3.43   |  0.039  | Significant main effect |
> | Sentiment       |  3.17   |  0.043  | Significant main effect |
> | Inspiration     |  2.96   |  0.027  | Significant main effect |
> | Overall         |  3.58   |  0.035  | Significant main effect |
>
> All **p-values < 0.05**, indicating significant differences among tutoring methods and confirming robust effects on subjective experience and learning outcomes.

---

### Official Review · Reviewer_8UDm · 2025-11-01

**Soundness:** 4
**Presentation:** 4
**Contribution:** 4
**Rating:** 8
**Confidence:** 4

**Summary:**

The authors introduce PELICAN, a large language model–based tutoring framework to deliver personalized education. It is designed to give adaptive feedback to students. It has two primary components: cognitive diagnosis and adaptive tutoring. The cognitive diagnosis uses a hierarchy of concept knowledge graph and uses an expert–assistant–verifier to generate diagnostic questions. Adaptive tutoring creates explanations and strategies based on the diagnosed cognitive state. It uses a dual-system strategy: fast and slow thinking to tailor based on the necessity of student. By extensive experiments across synthetic and real students, the authors show that PELICAN can be highly effective for LLM-driven personalized tutoring that mirrors human pedagogical behavior.

**Strengths:**

The primary strength of the paper lies in the clarity of its presentation. The authors take an intuitive idea to adapt LLMs to humans and show such system can work in practice. The two major components in PELICAN have been clearly written and motivated. The authors explain the importance of each and every component in the training pipeline. For example, the cognitive modeling component uses a hierarchical knowledge tree, with a well-defined curriculum that traverses from the leaf to the root. In order to ensure robust question quality during diagnosis, Expert–Assistant–Verifier Pipeline. Finally, the authors explain the importance of both short and fast thinking in adaptive tutoring and how they are decided based on the diagnosed cognitive state. By clearly showing improvements on both synthetic and real world students, the authors show the efficacy of their proposed tutoring sysem.

**Weaknesses:**

As such, I don't have many concerns with the paper. Please find some of my questions regarding the experiment setup.

a) How fine-grained are the knowledge concepts in the hierarchy? Furthermore, does the system allow more granularity in the concepts, depending on the student?

b) The expert-assistant-verifier pipeline depends on the accuracy of the two LLMs involved. Do the authors conduct ablations on what fraction of the diagnostic questions are noisy? And how much is the performance of PELICAN affected by the noise?


Furthermore, the authors should discuss on the diversity of the students involved in the human study. More than just performing well, did the students also report favorable experiences when interacting with LLMs? For example, a hyperparameter that can be controlled is M, which controls the number of times the LLM switches from fast to slow thinking. Is M decided based on whether the student progresses in the problem?

**Questions:**

Please see above for my questions.

---

> ### Author Response · Authors · 2025-11-23
>
> ## Q1 Granularity of Knowledge Points
>
> To address the reviewer’s question, we provide the following clarification with examples:
>
> (1) example 1:
>
> ├── Inheritance
>
> │   ├── Gene Mutation
>
> │    │   ├── Base Substitution, Insertion, and Deletion
>
> │    │    │    └── Allelic Variation of Genes
>
> (2) example 2:
>
> ├── Complex Numbers
>
> │   ├── Conjugates of Complex Numbers
>
> │   ├── Modulus of Complex Numbers
>
> │   └── Multiplication of Complex Numbers
>
> (3) example 3:
>
> ├── Organic Chemistry
>
> │   ├── Isomers
>
> │   │   ├── Chain Isomers
>
> │   │   └── Functional Group Position Isomers
>
> In practical applications, the granularity of concepts within the knowledge hierarchy is typically standardized by experts based on the guidelines of the Gaokao, China’s most comprehensive secondary-education exam with broad coverage, a well-structured system, and an appropriate level of depth. Therefore, our current system employs a fixed-granularity knowledge tree structure to ensure consistency and alignment with these standards, providing equal treatment for all students.
>
> That said, we agree that dynamically adjusting concept granularity for individual students is a promising direction for future work. In principle, a student-specific concept structure could be constructed through a top-down and bottom-up diagnostic process, where coarse concepts are refined when necessary, and fine-grained nodes are merged when the student demonstrates consistent mastery. In the current system, the adaptive tutoring phase already supports a question-decomposition strategy, allowing instructors to break down problems into finer-grained sub-concepts as needed. This ensures flexibility in addressing individual student needs even with the fixed granularity.
> ## Q2 Noise in the Pipeline
>
> **Noise Ratio** In Table 1, we compare the diagnostic accuracy of the PELICAN method (94.93%) with that of the No-pipeline model (93.92%). To quantify the proportion of noise, we examined the diagnostic questions generated by both methods. The error types can be categorized into: (a) question generation errors, (b) verifier misjudgments, (c) inconsistencies between student responses and cognitive states, and (d) misdiagnosis caused by dependency relations.
>
> |             |  a   |  b   |  c   |  d   |
> | :---------- | :--: | :--: | :--: | :--: |
> | No-pipeline | 2.6  | 0.8  | 2.1  | 0.6  |
> | PELICAN     | 1.7  | 0.9  | 2.0  | 0.5  |
>
> **Noise Impact** We used 49 math problems during the second-stage tutoring to evaluate the impact of noise in both methods:
>
> | Methods     | R_coverage | F_frequency | Overall | Avg_round |
> | ----------- | -------------------- | --------------------- | ------- | --------- |
> | No-pipeline | 72.83                | 72.92                 | 4.22    | 8.93      |
> | PELICAN     | 73.09                | 73.15                 | 4.28    | 8.56      |
>
> The findings show that lower diagnostic accuracy in the first stage slightly affects second-stage tutoring focus and increases tutoring rounds. For example, one misdiagnosis led the teacher to incorrectly focus on explaining the concept of sets, resulting in unnecessary additional tutoring.
> ## Q3 Tutoring Experience
>
> **Student Feedback** At the end of the experiment, students completed a satisfaction survey and highlighted the most helpful dialogue turns. Feedback showed benefits beyond task performance:
>
> - 34% felt more at ease interacting with the virtual teacher;
> - 27% found the guidance targeted;
> - Among low-performing students, 39% considered examples and step-by-step explanations most helpful.
>
> Students marked 13% of key turns in the fast-thinking phase and 24% in the slow-thinking phase, indicating stronger preference for the slow-thinking phase.
>
> **Role of M**
>
> The hyperparameter M triggers a more deliberate tutoring strategy when a student struggles with a concept over multiple rounds. Testing M = 1, 2, 3 showed that M = 2 provides the best balance of learning outcomes, efficiency, and token cost.
> The slow-thinking phase uses only 24% of tokens, suggesting that M = 2 already achieves a good trade-off. Further optimization could adjust switching based on how similar the confusion is across rounds—switching earlier when confusion is similar, and later when it differs, which may indicate exploration of new knowledge points, since the fast-thinking phase has not yet encountered a bottleneck.

---

### Author Response · Authors · 2025-12-03

We sincerely thank the reviewers for their time and feedback. We are grateful for their recognition during the rebuttal, which raised our score from **2468** to **2668**. Importantly, our rebuttal process and score improvement (Nov. 25) occurred before the bug disclosure on Nov. 27, and no information was leaked. We are encouraged by the positive reception of our work, particularly in the following areas:

- **Novel and Sound Framework Design** (8UDm, xBJb, 5Chj, eUTL): Reviewers commended the well-motivated framework, specifically highlighting the "fast and slow thinking" dual-system strategy as a compelling and rational design choice.

- **Comprehensive and Systematic Evaluation** (8UDm, 5Chj, eUTL): The systematic experiments on synthetic and real-world benchmarks were praised for effectively demonstrating the method's superiority through comprehensive comparisons.

- **Clear Presentation and Quality Writing** (8UDm, xBJb, 5Chj, eUTL): The high-quality writing and logical structure of the paper were appreciated, with reviewers noting that the intuitive explanations and clear figures facilitated ease of understanding.

During the rebuttal period, we undertook substantial efforts to address the reviewers’ comments and concerns. The key updates to the paper include:

- **Expanded User Study**: To strengthen statistical robustness, we increased the number of participants in our real-world experiment **from 67 to 169 students**. A **power analysis** was conducted to verify the adequacy of this sample size. Additionally, **ANOVA tests** were performed to confirm statistically significant improvements across key metrics, providing stronger evidence for the framework’s effectiveness.
- **Introduction of Quantitative Metrics**: To reduce reliance on subjective evaluations, we introduced new metrics, R_coverage and F_frequency, to objectively assess tutoring effectiveness.
- **Clarification of Scope and Contributions**: Our work is the first to integrate online cognitive diagnosis with adaptive tutoring into a unified system, wherein diagnosis drives tutoring, and tutoring subsequently refines diagnosis. This approach addresses a longstanding challenge in AI-driven education: **bridging the gap between assessment and intervention**. Moreover, the framework offers valuable insights for other diagnostic domains such as psychology and medical counseling. Additionally, the inclusive ICLR community recognizes both theoretical advancements and practical applications, **similar application-driven agent research has also been accepted by the ICLR community**.

We sincerely apologize for the delayed response, which was caused by the need to coordinate student schedules for real-world experiments and the overlap with the recent OpenReview security incident that disrupted submissions and reviewer discussions. Despite these challenges, we conducted rigorous experiments to address the reviewers’ concerns. We hope the Area Chair will consider these exceptional circumstances when evaluating our submission.

---

### Meta-Review · Area_Chair_69zt · 2025-12-11

**Summary:**

The authors present an LLM-based adaptive tutoring system. The presentation is very applied and without the appendix not self-contained and hence remains often vague; alternatives for the design choices do not become clear and the reader often has to believe the text. This impression is also summarised by the more critical reviewers. In addition, there's only a very few equations that could provide rigor to the otherwise purely textual presentation, however, there appear to be many more variables than equations and their definition is often far away (e.g., 1-2 pages) from their first occurrence in an equation. This renders understanding difficult and it does not become clear why we need to introduce the variables and equation at all. Technically, there does not seem to be much novelty but again, this could be hidden well in the text. I am under the impression that the presentation is suboptimal for ICLR and better suited for a dedicated conference on educational research. The paper might hide some interesting ideas but is not ready for publication at ICLR at the current stage.

**Reviewer Concerns:**

Reviewers are very different opinions on the paper. However, most of the reviewers have major concerns with the paper indicated by many "poor" ratings. Given the way of presentation, the paper addresses a very different and much more applied target audience than ICLR reviewers. Consequentially, they ask for more rigor in the presentation and technical details that one could use as an anchor for discussion. It is very difficult to convey technical advances without equation as all math has to be explained in text. Thus, I believe that, although the authors responded extensively to the reviewers, the major concerns could not have been resolved in the rebuttal.

**Reviewer Scores:**

The authors did very well in providing additional details and also experimental results, references etc. The core problem of the paper however seems to be the textual writeup that leaves too many open questions. In such a case,  it is nearly impossible for reviewers to list all issues and at the same time for authors to address all these issues in a rebuttal. The paper needs a major rewriting to incorporate rigor and math at all stages of the writeup to possibly satisfy all questions and comments. Hence, I don't believe that reviewers would have changed their opinion on the paper based on the (thorough) responses by the authors.

---

### Decision · Program_Chairs · 2026-01-26

Reject